# Outlier-Free SpeechLM for Fast Adaptation and Robust Quantization

## Abstract

We introduce SOFA (Stabilized Outlier-Free Attention), a drop-in replacement for the softmax activation that tackles the attention-outlier problem when turning a text-only LLM into a speech-text multi-modal model (SpeechLM). Our primary observation is that outliers emerge from both *multi-modal low-rank adaptation* and *post-training quantization* of transformer attention, degrading state-of-the-art SpeechLMs performance. To address these issues, we leverage a pretrained language model as a foundation and replace the standard softmax attention with SOFA which can be applied as a drop-in replacement of the vanilla softmax. We propose a plug-in method that directly eliminates outliers without adjusting pretraining weights and quantitatively measure the prevalence and impact of outliers in a unified speech-text transformer. We evaluate two multi-modal adaptation strategies: full fine-tuning on multi-modal data followed by post-training quantization, and apply LoRA on SOFA equipped model (SOFA-LoRA adapter) which keeps the pretrained LLM frozen without extra pre-training. The full fine-tuning route delivers strong, consistent gains across all modalities (textLM, SpeechLM, ASR, TTS), whereas the SOFA-LoRA adapter without touching any pretrained weights—surpasses the vanilla-LoRA adapter baseline and is particularly effective on text-output tasks such as ASR, all while retaining full compatibility with standard LLM checkpoints. Empirically, on the OPT-1.3b model, incorporating SOFA into SpeechLM yields a 88% improvement in multi-modal low-rank adaptation and a 37% improvement in post-training quantization.

## 1 Introduction

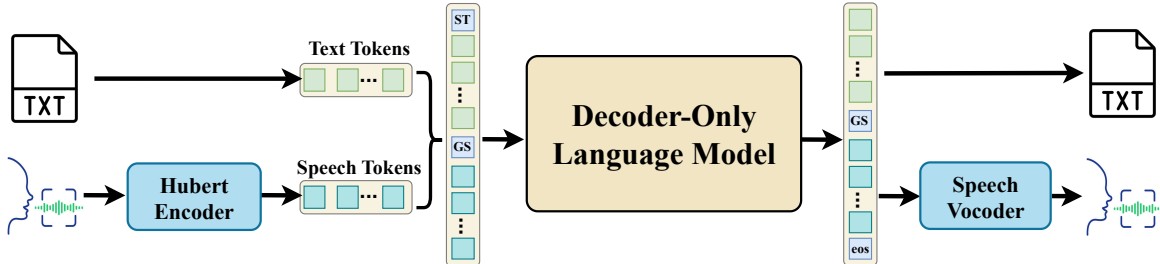

Figure 1: **Overview of the SpeechLM System.** The model is trained with a next-token prediction objective over a joint text and speech tokens vocabulary. Speech tokens are derived from continuous waveforms using a HuBERT encoder and can be decoded back into audible speech. We train on a mixture of sequences: text-only, speech-only, speech-to-text (ASR), and text-to-speech. Illustrated here is a TTS example, where special tokens "ST" (start of text) and "GS" (generate speech) indicate the task and desired output modality.

Recent efforts to adapt text-only Large Language Models (LLMs) for speech-text applications (SpeechLM) expose under attention-outlier problem: extreme values appears during low-rank multimodal adaptation and become even more damaging after post-training quantization (Wei et al., 2023; Bondarenko et al., 2023). These outliers slow convergence, destabilize training, and erase much of the accuracy promised by

parameter-efficient tuning such as LoRA (Huang et al., 2024; Chhabra et al., 2025). We tackle the bottleneck with Stabilized Outlier-Free Attention (**SOFA**), a drop-in replacement for the softmax activation that can be applied to pretrained transformers without re-training. By neutralizing outliers at their source, SOFA lets a text LLM adapt faster, survive 4-bit quantization, and surpass strong vanilla-softmax LoRA baselines on ASR, TTS, and other SpeechLM tasks under the same compute budget.

SpeechLM leverages pretrained language models to enhance speech recognition and synthesis (Nguyen et al., 2025), as in Figure 1. By employing a unified token space for both speech and text, it accommodates automatic speech recognition (ASR), text-to-speech (TTS), speech generation (SpeechLM), and text generation (textLM) tasks within a single framework (Maiti et al., 2024; Yang et al., 2024). This holistic design simplifies the training pipeline and fosters knowledge sharing among diverse applications.

Despite these advances, significant challenges persist. Many SpeechLMs cannot directly generate speech; they either only accept multi-modal inputs (Chu et al., 2023; Wang et al., 2023a; Gong et al., 2023; 2024) or require fully fine-tuning the entire model to handle both text and speech outputs (Maiti et al., 2024; Yang et al., 2024; Zhang et al., 2024; Défossez et al., 2024). Such large-scale full-model fine-tuning imposes substantial computational overhead, limiting widespread adoption. Moreover, post-training quantization (Xiao et al., 2023), a practical approach for deploying large models in resource-constrained settings, often suffers from outliers inherited from pretrained LLMs or introduced by multimodal data, thus impairing accuracy and diminishing the benefits of low-rank adaptation (Clark et al., 2019a; Kovaleva et al., 2019a; Zhao et al., 2024; Huang et al., 2024; Crabbé et al., 2024).

Attention outliers arise as a natural consequence of extending text-pretrained transformers to multi-modal settings (e.g., text and speech modality) under parameter-efficient adaptation. Speech tokens differ from text tokens in both statistical structure and semantic density. It often forms longer and more repetitive sequences derived from clustered acoustic units rather than linguistic subwords (Hsu et al., 2021; Zhang et al., 2024; Maiti et al., 2024). When injected into a text-pretrained attention space, this distributional mismatch causes certain tokens to receive disproportionately large attention logits. Moreover, low-rank adaptation methods such as LoRA constrain updates to a low-dimensional subspace, limiting the model's ability to smoothly redistribute attention mass across modalities (Hu et al., 2022; 2025; Huang et al., 2024). As a result, adaptation pressure is often absorbed by a small subset of attention entries, producing extreme values. These effects are further amplified by residual connections, which propagate early attention imbalances across layers (Clark et al., 2019b; Kovaleva et al., 2019b).

To tackle these outliers, we simply swap the standard self-attention mechanism (Vaswani et al., 2017) with SOFA, an outlier-free activation inspired by Hopfield methods (Hu et al., 2024a). This delivers two key benefits. First, SOFA avoids the infinite-loss pitfalls that can occur when training outlier-handling layers in large models (e.g., OPT-1.3b), leading to a more scalable and robust solution. Second, SOFA eliminates the need to pretrain a specialized outlier-resistant model from scratch by seamlessly integrating a dedicated stabilization module into an existing Large Language Model (LLM). This design preserves pretrained strengths from vanilla LLMs while outl outlier mixing that could degrade fine-tuning and quantization. Empirically, our method consistently outperforms standard transformer-based layers when combined with low-rank adaptation techniques such as LoRA (Hu et al., 2022) and post-training quantization (Xiao et al., 2023).

**Contributions.** We propose a simple, Stabilized Outlier-Free Attention (**SOFA**), to eliminate outliers in multi-modal speech-text transformers. Our key contributions are:

- **First Quantitative Outlier Analysis.** To our knowledge, we are the first to measure and characterize activation and attention outliers in a unified SpeechLM system, revealing their impact on cross-modal adaptation and quantization.

- **Adapt to pretrained LLM with low-cost fine-tuning.** We introduce SOFA, a new attention can be adapted on to pretrained LLM, improving down-stream task performance when appling LoRA without changing a lot model weight.

- **Unified Adaptation and Efficiency Gains.** We demonstrate that, across LoRA, QLoRA, and post-training quantization on the OPT family (Zhang et al., 2022), SOFA delivers an 88% improvement in

low-rank multi-modal adaptation and a 37% reduction in quantization degradation, all within the same compute budget.

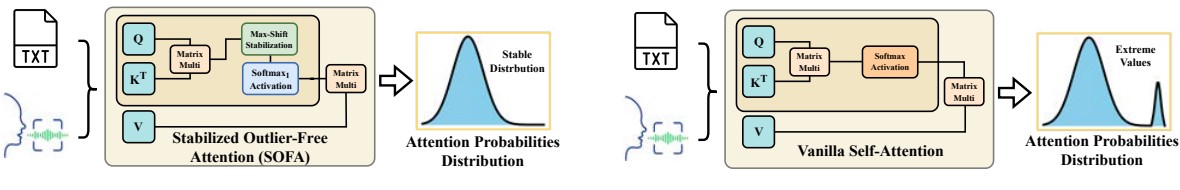

(a) SOFA Attention Mechanism.  (b) Vanilla Attention Mechanism.

Figure 2: **Comparison of Attention Mechanisms.** (a) The Stabilized Outlier-Free Attention (SOFA) mechanism introduces max-shift normalization and a specialized Softmax$_1$ activation function to reduce outliers in multimodal adaptation. (b) The vanilla attention mechanism, in contrast, often leads to significant output outliers during adaptation.

## 2 Related Work

**Discrete Speech Representation.** Recent advances in Self-Supervised Learning (SSL) for speech enables the extraction of meaningful representations from raw audio. Models like HuBERT (Hsu et al., 2021) and w2v-BERT (Chung et al., 2021) generate discrete speech tokens by clustering learned features, capturing the linguistic content of speech. This transforms speech into pseudo-text, facilitating applications in speech-based natural language understanding and generation. By clustering continuous features into discrete tokens representing phonetic or sub-word units, these models improve the accuracy and efficiency of tasks such as TTS (Hayashi and Watanabe, 2020), speech-to-speech translation (S2ST) (Lee et al., 2022), and ASR (Park et al., 2019). In addition to SSL-derived units, another line of research utilizes neural audio codec-based representations, such as Residual Vector Quantization (RVQ)-based codecs like EnCodec (Défossez et al.) or SoundStream (Zeghidour et al., 2022). These codecs produce acoustic-level tokens optimized for high-fidelity waveform reconstruction. While codec tokens often yield superior audio quality in generation tasks, they typically result in much longer sequences and distinct statistical distributions compared to text, posing unique challenges for unified speech–text modeling and quantization.

**Speech and Text LMs.** Joint modeling of speech and text has gained significant attention in recent studies. LLM-initial approaches (Ao et al., 2022; Chen et al., 2022) proposed learning shared speech-text representations with separate encoders and decoders, requiring alignment losses for multi-modal transfer. Recent methods employ a single model for multiple tasks. For example, SpeechGPT (Zhang et al., 2023) combines audio generation with textLMs, PolyVoice applies speechLM to S2ST (qian Dong et al., 2023), SpiritLM (Nguyen et al., 2025) excels in speech and expressive speech generation, also adapted for related speech tasks, and Voxtlm (Maiti et al., 2024) conducts speech/text generation along with ASR and TTS. In these textless NLP and unified frameworks, discrete units serve as the key interface enabling speech continuation and translation without explicit text supervision. We utilize textually pretrained OPT (Zhang et al., 2022) for better initialization inspired by (Maiti et al., 2024; Hassid et al., 2024) and leverage different speech tokens, ensuring reproducibility.

**Low-Rank Adaptation and Post Training Quantization.** Low-Rank Adaptation (Xin et al., 2024) and Post Training Quantization (PTQ) (Gholami et al., 2022) are essential techniques for reducing the memory footprint and latency of large foundation models (Bommasani et al., 2021), i.e. huge transformer-based models. Those large foundation models play a crucial role not only in machine learning area but also in a huge scientific area, such as (Zhou et al., 2025) for genomics, (Wang et al., 2023b; Wu et al., 2023) for financial, and (Maiti et al., 2024) for speech. However, large foundation models are resource-intensive. Low-Rank Adaptation and PTQ play crucial roles in deploying these large models on edge devices with limited resources. Significant contributions are made in the area of low-rank adaptation (Dettmers et al., 2024; Li et al., 2023; Hu et al., 2022) and PTQ (Zafrir et al., 2019; Dettmers et al., 2022). Despite significant contributions to low-rank adaptation and PTQ, substantial challenges still remain. Recently, several studies focus on mitigating the impact of outliers in model quantization (Bondarenko et al., 2024; Xiao et al., 2023). Additionally, the outlier challenge also influences both pre-training (Hu et al., 2024a) and fine-tuning (Chen

et al., 2024; Hu et al., 2025). However, none of these studies focus on low-rank adaptation. Our contribution is orthogonal to the specific choice of speech tokenization; we analyze how existing discrete representations interact with parameter-efficient fine-tuning. We specifically highlight how outlier activations can be amplified in discrete-token-based pipelines and demonstrate that SOFA effectively stabilizes these representations without modifying pretrained weights. To tackle this issue, we propose the Outlier-Free Layer to manage outliers effectively during both the low-rank adaptation and quantization processes.

## 3 Methodology

This section introduces our apporach for integrating Stabilized Outlier-Free Attention (SOFA) into a SpeechLM framework. By replacing the standard transformer attention mechanism with a stabilized, outlier-free variant, we address the challenges posed by outliers during multi-modal low-rank adaptation and post-training quantization. We begin by outlining the SpeechLM setup, then describe the outlier-free architecture and how it integrates into SpeechLM, and finally provide theoretical justifications for our design.

### 3.1 SpeechLM Setup

Our goal is to model speech and text modalities within a unified framework. To achieve this, we convert continuous speech signals into discrete tokens $s_i \in \mathcal{V}_{\text{sp}}$ via a speech tokenizer (e.g., a HuBERT-based model with $k$-means clustering). These speech tokens are integrated with text tokens $t_i \in \mathcal{V}_{\text{txt}}$ to form a joint vocabulary $\mathcal{V}_{\text{joint}} = \mathcal{V}_{\text{txt}} \cup \mathcal{V}_{\text{sp}}$. We train on a mixture of tasks (ASR, TTS, speech and text generation) using subword models (e.g., BPE or SentencePiece) for both text and speech tokens. This shared vocabulary allows the model to predict the next token directly, regardless of the modality. Concretely, we model the probability of a text utterance $T = (t_i)$ as $p(T) = \prod_i p(t_i \mid t_1, \cdots, t_{i-1})$, and similarly represent continuous speech signals as discrete tokens $S = (s_i)$, which are modeled in the same manner. The joint probability of speech and text tokens $J = (j_i \in \mathcal{V}_{\text{joint}})$ is expressed as $p(J) = \prod_i p(j_i \mid j_1, \cdots, j_{i-1})$. To handle multiple tasks in a framework, we apply four special tokens indicating the start and generation modes for speech or text, following (Maiti et al., 2024). For example, ASR sequences start with a "start of speech" token and use "generate text" as the prediction target, whereas TTS sequences start with a "start of text" token and use "generate speech" as the target. This unified autoregressive strategy simplifies handling ASR, TTS, speech generation (speechLM), and text generation (textLM) tasks within one model.

**Modality Fusion.** The joint vocabulary $\mathcal{V}_{\text{joint}}$ includes discrete speech tokens derived from clustered HuBERT features (Hsu et al., 2021). We employ HiFi-GAN (Kong et al., 2020) trained on LJSpeech (Ito., 2017) to synthesize speech waveforms from discrete tokens. We also apply SentencePiece (Kudo and Richardson, 2018) across both text and speech tokens, which reduces sequence lengths and provides richer contextual representations. During training, we use a teacher-forcing approach in an autoregressive manner. At each timestep $i$, the model predicts the distribution $\widehat{p}^i = \text{SpeechLM}(j_1, \ldots, j_{i-1})$., and we compute the cross-entropy loss as:

$$L_{CE}(p_i, \widehat{p}^i) = - \sum_{c=1}^{|\mathcal{V}_{\text{joint}}|} p_i(c) \log \widehat{p}^i(c),$$

where $p_i$ is the ground-truth distribution over tokens. At inference, the model predicts new tokens $\widehat{j_i}$ autoregressively, conditioned on the preceding context.

### 3.2 Outlier-Free Architecture

Modern multi-modal speech-text transformers often face severe outlier issues during both multi-modal low-rank adaptation and post-training quantization. Our Stabilized Outlier-Free Attention (SOFA) directly tackles these challenges, as illustrated in Figure 2.

**Problem Setup.** Consider an input sequence (speech and text) represented as a matrix $X \in \mathbb{R}^{d \times L}$, where $d$ is the feature dimension and $L$ is the sequence length. This input is fed into standard transformer layers composed of self-attention and feed-forward sub-layers.

**Attention Probability**     **Attention Output**

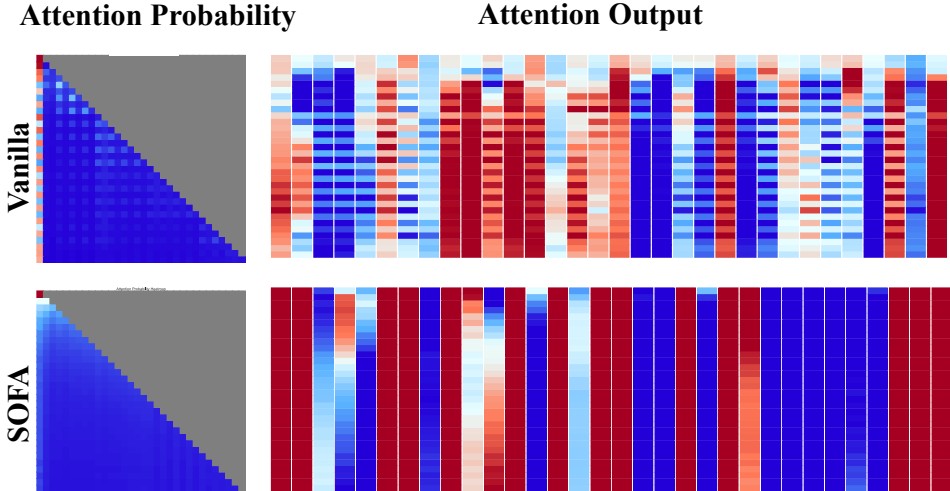

Figure 3: **Attention Visualization in LoRA Fine-Tuning.** Comparison of attention probability and output for a multi-modal speech sample in OPT-350m. Heatmaps from the last hidden layers are scaled from 0 (blue) to 1 (red). The vanilla model distributes attention broadly, diluting focus, while SOFA sharpens attention on key tokens, improving efficiency.

**Motivating Example.** Studies Clark et al. (2019a); Kovaleva et al. (2019a); Hu et al. (2024a) show that certain tokens with little information (e.g., punctuation) may receive disproportionately high attention weights, creating "outliers" that skew distributions. In a multi-modal context, these outliers often become more pronounced, worsening performance degradation during fine-tuning and quantization (Crabbé et al., 2024). We use this observation as a starting point by considering the following attention mechanism:

$$\text{Output} = \text{Residual}(\text{Softmax}(XW_q XW_k^{\mathbb{T}}/\sqrt{d})XW_v + X). \tag{3.1}$$

As shown in (Hu et al., 2024a), if the attention input $X$ already holds sufficient information, the attention mechanism within the residual connection should ideally act as an identity transform, producing near-zero attention weights ($\text{Softmax}(QK^{\text{T}})V$). In such cases, tokens with high values in $V$ should receive near-zero attention probabilities, as determined by $\text{Softmax}(QK^{\text{T}})$. However, the classic Softmax function normalizes probabilities in a way that disproportionately amplifies the attention probabilities assigned to low-value tokens. This broadens the distribution of attention scores and introduces outliers that degrade the model's performance. Additionally, integrating speech and text — two modalities with inherently different statistical properties — can further skew attention distributions.

**Our Approach: Stabilized Outlier-Free Attention (SOFA).** We propose SOFA as a attention mechanism to mitigate outlier effects in large-scale transformer models. Drawing on memory-associated activations (Miller, 2023; Hu et al., 2024a), SOFA replaces the standard softmax with an operation called $\text{Softmax}_1$, which specifically reduces the disproportionate influence of extreme values. The key steps are:

$$S \leftarrow S - \max(S), \tag{3.2}$$

$$\text{Softmax}_1(S) = \frac{\exp(S)}{1 + \sum_{i=1}^{L} \exp(S_i)}, \tag{3.3}$$

The first step, $S \leftarrow S - \max(S)$, constitutes our stabilized layer. By subtracting the largest element of $S$, often called a "max-shift", we avoid numerical overflow and normalize outlier values before applying $\text{Softmax}_1$. Similar to the stabilized method in (Dao et al., 2022), this design ensures that extremely large logits are brought to a manageable range, thereby curbing outlier formation in the attention weights. Although $\text{Softmax}_1$ mitigates outliers, it can introduce gradient instabilities in larger models (Alman and Song, 2024). Our stabilized layer (the max-shift) addresses this by keeping values within a stable numeric range, which preserves consistent against exploding gradients during fine-tuning or quantization (Hu et al., 2025; Jiang et al., 2023). Figure 3 illustrates how SOFA reshapes attention distributions during fine-tuning. We compare a vanilla transformer to one using SOFA by color-coding final hidden representations (red for higher values, blue

for lower). The vanilla model's attention is scattered accross many tokens, leading to inefficient computation and degraded performance.

### 3.3 Integrating SOFA into SpeechLM

We incorporate SOFA into a SpeechLM framework to effectively manage both speech and text modalities. This integration aim to leverage pretrained LLM parameters and supports robust post-training quantization, but it presents two main challenges:

**Multi-modal Adaptation.** Extending a text-only LLM to multi-modal tasks is nontrivial because speech tokens greatly expand the vocabulary $\mathcal{V}_{\text{joint}}$ and typically increase sequence lengths fivefold. Together with preexisting outliers, these changes complicate typical parameter-efficient training or quantization. SOFA addresses this with the $\text{Softmax}_1$ function (Equation (3.3)) to avoid outliers emerging, stabilizing attention distributions under increased vocabulary sizes and multi-modal data. This makes fine-tuning with LoRA or performing post-training quantization (e.g., SmoothQuant) more feasible, reducing the computational overhead of full-model tuning (Maiti et al., 2024; Nguyen et al., 2025) while improving final performance.

**Stabilized Outlier-Free Adaptation.** Simply substituting softmax-based attention with SOFA can create parameter mismatches, since the original weights were tuned for a vanilla softmax function. Our stabilized layer in Equation (3.2) ensures consistent gradients, allowing direct initialization from off-the-shelf LLM parameters—no specialized outlier-free pretraining is necessary. As a result, SpeechLM equipped with SOFA achieves stable and efficient adaptation to new tasks and modalities, demonstrating substantial gains in both ASR and TTS compared to vanilla transformer-based SpeechLM systems.

In practice, the SOFA-LoRA adapter preserves the original language capabilities of the underlying LLM, inherits SOFA's robustness to activation outliers, and adds only a tiny fraction of trainable parameters. Applied to SpeechLM, this configuration consistently outperforms the vanilla transformer baseline on both ASR and TTS benchmarks while tuning no more than two percent of the total model parameters, confirming that SOFA integrates with modern low-rank adaptation techniques.

### 3.4 Theoretical Justifications

Building on the theoretical advantages of the outlier-free transformer reported by (Hu et al., 2024a), we provide two additional justifications for applying LoRA to the outlier-free transformer.

**Expressiveness.** We emphasize that our design choices offer strong expressive guarantee for model expressiveness; specifically, Low-Rank Adaptation with $\text{Softmax}_1$ enhances the model's expressiveness, as demonstrated in Luo et al. (2025, Theorem A.2).

**Training Efficiency.** We find that the attention weights are concentrated on significant tokens, enabling less training time cost during fine-tuning compared to the vanilla version. We provide a theoretical justification for why we observe improved LoRA training efficiency.

**Proposition 3.1** (Fast LoRA Requires Proper Normalization (Informal Version of Proposition B.1)). Let $X \in \mathbb{R}^{d \times L}$ be the input sequence, and let $r$ denote the rank of the LoRA adapters for a pretrained transformer model. Sub-quadratic time-efficient LoRA training up to a precision of $\epsilon = O((\log L)^{-4})$ is achievable if the following conditions hold: (i) long sequence setting with $d = O(\log L)$, (ii) mild rank $r < d$, and (iii) proper normalization of the input and model weights.

**Remark 3.1.** Our outlier-free layer ensures proper normalization of the model weights. Our stabilization technique ensures proper normalization of the model input.

We defer detailed theoretical justifications to Appendix B.

## 4 Experimental Studies

In this section, we present experiments to assess the effectiveness of our proposed framework, benchmarking it against state-of-the-art methods from (Maiti et al., 2024).

Table 1: **Comparing SOFA with Vanilla Transformer in a Post-Training Quantization (PTQ) setting.** We conduct experiments across three quantization methods (SmoothQuant, AffineQuant, OmniQuant, SpQR) on a low bit weight and activation quantization setting – weight 4 bits and activation 4 bits (W4A4). Evaluation metrics include Text PPL, SpeechLM PPL, ASR WER, and TTS CER. We assess the average performance drop across these four tasks post-quantization. Results show SOFA consistently outperforms vanilla transformer, exhibiting smaller performance drops when applying low bit quantization methods, demonstrating its superior efficiency in PTQ settings.

| Model | Method | #Bits | Quantization Method | TextLM PPL (↓) | SpeechLM PPL (↓) | ASR WER (↓) | TTS CER (↓) | Avg Performance Drop (↓) |
|---|---|---|---|---|---|---|---|---|
| OPT-350m | Vanilla | W16/A16 | - | 13.13 | 43.10 | 8.42 | 17.56 | - |
| | | W4/A4 | SmoothQuant | 36.74 | 75.38 | 40.17 | 70.53 | 211.19% |
| | | W4/A4 | AffineQuant | 27.28 | 66.31 | 36.84 | 40.83 | 138.80% |
| | | W4/A4 | OmniQuant | 27.85 | 67.83 | 37.54 | 41.37 | 143.34% |
| | | W4/A4 | SpQR | 29.36 | 69.28 | 38.57 | 42.46 | 171.06% |
| | SOFA | W16/A16 | - | 13.47 | 43.34 | 9.81 | 17.31 | - |
| | | W4/A4 | SmoothQuant | 23.48 | 62.17 | 36.22 | 40.83 | **116.02%** |
| | | W4/A4 | AffineQuant | 22.82 | 51.74 | 25.78 | 28.44 | **71.90%** |
| | | W4/A4 | OmniQuant | 22.83 | 52.08 | 26.11 | 29.15 | **73.82%** |
| | | W4/A4 | SpQR | 22.88 | 52.78 | 27.02 | 29.11 | **83.81%** |
| OPT-1.3b | Vanilla | W16/A16 | - | 12.62 | 41.33 | 8.00 | 18.73 | - |
| | | W4/A4 | SmoothQuant | 36.74 | 87.46 | 48.96 | 53.15 | 221.61% |
| | | W4/A4 | AffineQuant | 24.31 | 61.74 | 43.68 | 32.47 | 128.45% |
| | | W4/A4 | OmniQuant | 24.43 | 62.38 | 44.52 | 33.03 | 131.40% |
| | | W4/A4 | SpQR | 25.85 | 63.36 | 45.28 | 36.14 | 179.27% |
| | SOFA | W16/A16 | - | 12.95 | 42.48 | 8.25 | 12.07 | - |
| | | W4/A4 | SmoothQuant | 23.83 | 58.33 | 32.27 | 33.12 | **128.20%** |
| | | W4/A4 | AffineQuant | 20.81 | 48.84 | 22.78 | 25.46 | **81.12%** |
| | | W4/A4 | OmniQuant | 20.88 | 48.97 | 23.58 | 26.83 | **85.38%** |
| | | W4/A4 | SpQR | 22.74 | 49.42 | 25.36 | 27.13 | **106.03%** |
| Qwen2.5-7b | Vanilla | W16/A16 | - | 10.15 | 38.62 | 10.54 | 11.48 | - |
| | | W4/A4 | SmoothQuant | 18.93 | 60.24 | 43.22 | 35.28 | 164.98% |
| | | W4/A4 | AffineQuant | 16.54 | 56.72 | 30.77 | 34.18 | 124.89% |
| | | W4/A4 | OmniQuant | 16.23 | 55.93 | 29.28 | 33.17 | 117.88% |
| | | W4/A4 | SpQR | 16.78 | 57.02 | 31.12 | 35.88 | 130.21% |
| | SOFA | W16/A16 | - | 8.76 | 32.51 | 8.34 | 11.25 | - |
| | | W4/A4 | SmoothQuant | 14.77 | 52.18 | 27.98 | 29.78 | **132.36%** |
| | | W4/A4 | AffineQuant | 12.12 | 46.54 | 18.92 | 23.74 | **79.87%** |
| | | W4/A4 | OmniQuant | 11.96 | 45.33 | 17.61 | 20.22 | **66.74%** |
| | | W4/A4 | SpQR | 12.35 | 46.89 | 19.15 | 24.08 | **82.25%** |

**Models.** We evaluate two sizes of OPT models, OPT-350m and OPT-1.3b, to demonstrate scalability across model capacity. To further assess robustness at larger scale and across model families, we additionally evaluate our method on a recent backbone, Qwen2.5-7b. We adopt HuBERT k-means with $k = 200$ to produce discrete speech tokens and employ a shared BPE tokenizer across TextLM, SpeechLM, ASR, and TTS tasks.

**Datasets.** We employ four datasets across various tasks. For textLM, we use Librispeech (Panayotov et al., 2015), consisting of 40 million text utterances. SpeechLM employs LibriLight (LL) (Kahn et al., 2020), which contains 60,000 hours of audiobook recordings from 7,000 speakers, totaling 12 million utterances. For ASR, we use the English Multilingual Librispeech (MLS) dataset (Pratap et al., 2020). TTS experiments are conducted on LibriTTS (LT) (Zen et al., 2019) and VCTK (VC) (Veaux et al., 2017) datasets.

**Evaluation Metrics.** We adopt task-specific metrics to evaluate performance. We use perplexity (PPL) on models sharing identical vocabulary sizes for speech and text generation. ASR performance is measured via word error rate (WER). For TTS, Hifi-GAN (Kong et al., 2020) serves as the vocoder, and intelligibility is gauged through character error rate (CER) obtained from Whisper (Radford et al., 2023) decoding. We use geometric mean to demonstrate the average performance drop across all tasks. Lower scores in these

Table 2: **Comparison of SOFA-Based Transformer and the Vanilla Transformer under Low-Rank Adaptation.** We evaluate SOFA and vanilla attention across two Low-Rank Adaptation methods (LoRA and QLoRA), including SmoothQuant (SQ) with 8-bit and 4-bit precision. Metrics evaluated are Text PPL, SpeechLM PPL, and WER for ASR. In this setting, we push both SOFA-LoRA and vanilla LoRA to the extreme, by fine-tuning them across all four tasks. Because LoRA-based methods produce extremely high CERs for TTS, we omit TTS results here (see Section 4.3 for details). We also report the average performance drop after low-rank adaptation to assess SOFA's efficiency. Overall, SOFA achieves superior fine-tuning performance compared to the vanilla transformer.

| Model | Method | Low-Rank Adaptation Method | TextLM PPL ($\downarrow$) | SpeechLM PPL ($\downarrow$) | ASR WER ($\downarrow$) | Average Performance Drop ($\downarrow$) |
|---|---|---|---|---|---|---|
| OPT-350m | Vanilla | Full | 13.13 | 43.10 | 8.42 | - |
| | | LoRA | 17.87 | 51.65 | 93.91 | 163.00% |
| | | LoRA+SQ (8Bits) | 20.54 | 56.88 | 95.36 | 185.95% |
| | | LoRA+SQ (4Bits) | 27.31 | 60.03 | 97.24 | 221.17% |
| | SOFA | Full | 13.47 | 43.34 | 9.81 | - |
| | | LoRA | 17.71 | 51.13 | 18.52 | **40.95%** |
| | | LoRA+SQ (8Bits) | 18.91 | 53.24 | 25.47 | **64.82%** |
| | | LoRA+SQ (4Bits) | 25.89 | 59.11 | 28.32 | **96.33%** |
| OPT-1.3b | Vanilla | Full | 12.62 | 41.33 | 8.00 | - |
| | | LoRA | 17.14 | 50.22 | 46.92 | 113.13% |
| | | LoRA+SQ (8Bits) | 20.25 | 56.13 | 48.14 | 135.81% |
| | | LoRA+SQ (4Bits) | 27.01 | 59.28 | 60.21 | 184.81% |
| | SOFA | Full | 12.95 | 42.48 | 8.20 | - |
| | | LoRA | 16.83 | 49.51 | 8.26 | **14.89%** |
| | | LoRA+SQ (8Bits) | 18.11 | 52.68 | 10.83 | **31.55%** |
| | | LoRA+SQ (4Bits) | 24.73 | 58.47 | 16.47 | **73.77%** |

metrics denote better performance. Besides, we report next-token accuracy in the ablation study (Section 4.4) to assess model effectiveness. All reported results are averaged over three independent runs with different random seeds. The resulting standard deviations are consistently below 0.2% and are omitted for brevity.

## 4.1 Computational Resource.

All experiments are conducted on four NVIDIA A100 GPUs (80GB) and a 24-core Intel(R) Xeon(R) Gold 6338 CPU (2.00GHz). Our implementation is built in PyTorch and utilizes the HuggingFace Transformer Library.

## 4.2 Post-Training Quantization (PTQ)

To evaluate the efficiency of our method, we replace the standard attention layer (Vaswani et al., 2017) in all OPT models (Zhang et al., 2022) with our proposed Stabilized Outlier-Free Attention (SOFA). We then fine-tune these pre-trained OPT checkpoints at full rank, following (Maiti et al., 2024) but applying the SOFA-based transformers. Afterward, we evaluate the test sets in FP16 (16-bit floating-point) and apply state-of-the-art PTQ methods to gauge the performance drop from FP16.

**Baselines.** We use the model architecture from (Maiti et al., 2024) as the baseline for speech-text tasks. To evaluate quantization performance, we apply three state-of-the-art methods—SmoothQuant (Xiao et al., 2023), AffineQuant (Ma et al., 2024), OmniQuant (Shao et al., 2024) and SpQR (Dettmers et al., 2023)—to both the baseline model and our SOFA-based transformer. We adopt the hyperparameters recommended in each quantization study for consistency and fair comparisons.

**Results.** Table 1 shows that SOFA outperforms standard training frameworks in W4A4 (Weight-4-bit, Activation-4-bit) post-training quantization. The vanilla transformer suffers significant performance degradation with low-bit quantization (W4/A4). For example, with AffineQuant, the vanilla transformer undergoes

performance drops of 138.80%, and 128.45% for OPT-350m and OPT-1.3b, respectively. In contrast, the SOFA-based transformer reduces these declines to 71.90% and 81.12%. For OPT-1.3b, SOFA achieves a 37% relative improvement in average W4A4 quantization performance, underscoring its robustness in low-bit quantization for large models. The benefits of SOFA become even more pronounced on the large-scale Qwen2.5-7b model. Under W4A4 quantization, the vanilla transformer experiences dramatic performance degradation, with average drops of 164.98% (SmoothQuant), 124.89% (AffineQuant), 117.88% (OmniQuant), and 130.21% (SpQR). By contrast, SOFA consistently yields much smaller degradations of 132.36%, 79.87%, 66.74%, and 82.25% under the same respective quantization methods. These results indicate that SOFA's outlier-free attention mechanism scales with model size and is effective in stabilizing extremely low-bit quantization for large language models. Additional W8A8 (Weight-8-bit, Activation-8-bit) results are presented in Appendix C.

### 4.3 SOFA Low-Rank Adaptation

To verify that swapping softmax with **SOFA** benefits parameter-efficient multi-modal adaptation while keeping all pretrained weights frozen, we compare SOFA against the vanilla transformer in two popular PEFT settings, LoRA and QLoRA, followed by SmoothQuant (SQ) at 8- and 4-bit precision.

#### 4.3.1 SOFA-LoRA Adapter for Speech and Text Generation

In our first SOFA-LoRA experiment group, we use the same setup as the full fine-tuning in Table 1, training jointly on TextLM, SpeechLM, ASR, and TTS. While prior Audio LLM studies typically apply LoRA only to the audio-understanding task (Tang et al.; Chu et al., 2023; Gong et al., 2023; Hu et al., 2024b), here we push both SOFA-LoRA and vanilla LoRA to the extreme by evaluating them across all four tasks.

**SOFA-LoRA Adapter Methods.** We evaluate SOFA and the vanilla transformer using LoRA (Hu et al., 2022) and QLoRA (Dettmers et al., 2024), alongside a full-rank baseline as in Table 1. Specifically, we adopt a rank of 256 and an alpha value of 256 for LoRA. For QLoRA, we keep these settings but substitute Int8 (Dettmers et al., 2022) quantized in place of the 4-bit NormalFloat (NF4) (Dettmers et al., 2024).

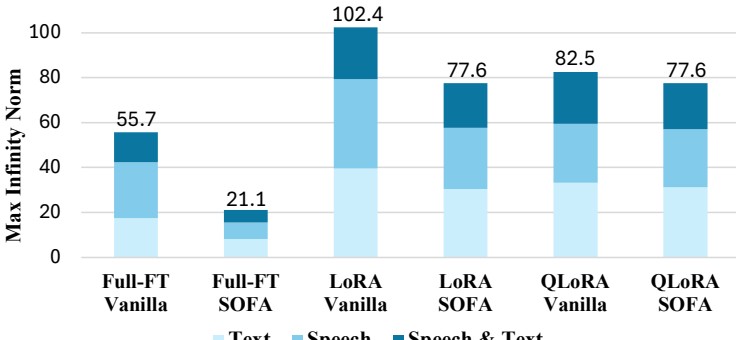

Figure 4: **Outlier Comparison Between SOFA-based Transformer and Baseline.** Comparison of the maximum infinity norm $\|\mathbf{x}\|_\infty$ across Text, Speech, and Speech+Text for SOFA-based transformer the vanilla transformer baseline. SOFA consistently shows lower $\|\mathbf{x}\|_\infty$ values across FT, LoRA, and QLoRA, indicating better outlier control than the baseline methods.

**Results and Observations.** Table 2 compares SOFA and the vanilla transformer baseline under two low-rank adaptation methods (LoRA and QLoRA), including 8-bit and 4-bit SmoothQuant (SQ). We report perplexity (PPL) on text (TextLM) and speech (SpeechLM), as well as Word Error Rate (WER) on ASR. TTS results are omitted due to the unsatisfactory performance of LoRA-based fine-tuning, which aligns with (Hao et al., 2025).

Despite excluding TTS, SOFA consistently outperforms the vanilla transformer across all tasks and quantization levels. For OPT-350m, the baseline approach suffers large performance drops with QLoRA (145%) and LoRA+SQ (4 bits, 221%), whereas SOFA reduces these drops to 46% and 96%, respectively. Similarly, for OPT-1.3b, the baseline QLoRA configuration incurs a 168% drop, and LoRA+SQ (4 bits) results in 185%. Under SOFA, these drops are reduced to 104% and 73.8%. One standout result is the non-quantized LoRA setting on OPT-1.3b: while the baseline experiences a 113% performance drop, SOFA brings it down to 14.9%, demonstrating SOFA's remarkable adaptability in low-rank scenarios. Overall, these results suggest that while

LoRA-based approaches remain challenging for TTS (requiring more extensive parameter updates (Maiti et al., 2024; Zhang et al., 2024; Défossez et al., 2024)), SOFA provides a robust and efficient fine-tuning framework for multi-modal text and speech understanding tasks.

### 4.3.2 Single-Task LoRA for Speech Recognition and Synthesis

Results in Table 2 indicate that jointly sharing a single low-rank adapter across TextLM, SpeechLM, ASR, and TTS is overly restrictive for LoRA-based adaptation. In particular, while SOFA mitigates the degradation observed in text and speech understanding tasks, TTS remains unstable under the multi-task LoRA setting, suggesting that speech generation requires more task-specific adaptation capacity. To disentangle the effect of task interference from the intrinsic limitations of LoRA, we conduct an additional controlled study where the model is fine-tuned with LoRA on a single task only. Specifically, we separately fine-tune the model for ASR-only and TTS-only training, while keeping all other pretrained parameters frozen and using identical LoRA hyperparameters as in Section 4.3.

Table 3 reports the results for ASR and TTS across three backbone sizes: OPT-350m, OPT-1.3b, and Qwen2.5-0.5b. Under this single-task setting, vanilla LoRA already yields reasonable performance, confirming that the degradation observed in the multi-task setup primarily arises from cross-task interference rather than an inherent inability of LoRA to adapt speech models. Importantly, SOFA consistently improves over the vanilla transformer across all model scales for both ASR and TTS. For ASR, SOFA reduces WER from 17.83% to 13.57% on OPT-350m and from 9.02% to 7.55% on OPT-1.3b. For TTS, SOFA also yields consistent CER reductions under single-task LoRA, decreasing CER from 15.09% to 14.62% on OPT-350m and from 14.31% to 12.57% on Qwen2.5-0.5b. These results indicate that once task interference is removed, SOFA remains effective even for speech generation under low-rank adaptation, improving stability and alignment despite the limited adaptation capacity of LoRA. SOFA complements LoRA by stabilizing attention dynamics, yielding robust gains whenever low-rank adaptation is applied in a well-scoped task setting.

### 4.4 Ablation Study

This section presents results from three perspectives. First, we analyze outlier differences between the vanilla and SOFA-based transformers. Then, we compare SOFA with some alternative techniques (Clipped Softmax and Gated Attention). Next, we explore how our training methodology contributes to efficiency. Finally, we evaluate SOFA across diverse settings, including the effect of weight clipping in PTQ, its performance in QLoRA under different quantization levels, and the standalone influence of the stabilization module on a standard Transformer.

**Quantitative Measurement about Outliers in the SpeechLM System.** Our quantitative assessment compares the vanilla transformer and SOFA using the maximum infinity norm $\|\mathbf{x}\|_\infty$ across three tasks and three low-rank adaptation methods. This metric strongly correlate with a model's robustness against outliers (Bondarenko et al., 2021; Hu et al., 2024a). Figure 4 shows that SOFA significantly reduces the max inf norm compared to vanilla method across all fine-tuning methods (Full-FT, LoRA, and QLoRA) and tasks. In the Speech task, SOFA achieves a 70% reduction in the max inf norm under Full-FT (24.95 to 7.46) and 23% under LoRA (39.82 to 27.27), highlighting its ability to suppress outliers. These improvements demonstrate that SOFA stabilizes activations, enabling better performance and robustness, especially under low-bit quantization settings, where vanilla method suffers significant degradation due to outlier instability.

Table 3: **Single-task LoRA fine-tuning for ASR and TTS.** We fine-tune the model using LoRA on a single task only (ASR-only or TTS-only), while keeping all pretrained parameters frozen and using the same LoRA hyperparameters as in the multi-task setting. Compared to joint multi-task LoRA (Table 2), single-task training significantly stabilizes both ASR and TTS. SOFA consistently outperforms the vanilla transformer across all backbone sizes, demonstrating its effectiveness under task-isolated low-rank adaptation.

| Model | Method | ASR WER ↓ | TTS CER ↓ |
|---|---|---|---|
| OPT-350m | Vanilla | 17.83 | 15.09 |
| OPT-350m | SOFA | **13.57** | **14.62** |
| OPT-1.3b | Vanilla | 9.02 | 11.08 |
| OPT-1.3b | SOFA | **7.55** | **10.47** |
| Qwen2.5-0.5b | Vanilla | 12.51 | 14.31 |
| Qwen2.5-0.5b | SOFA | **11.56** | **12.57** |

Table 4: **Comparison of SOFA with alternative attention modifications under various quantization settings.** At 8-bit quantization, all methods show similar performance (<0.5% drop). However, under 4-bit quantization, SOFA outperforms Clipped Softmax, Gated Attention, and vanilla approaches, illustrating its robustness in extreme quantization scenarios.

| Model | Method | W/A Bits | Text PPL ↓ | Speech PPL ↓ | ASR WER ↓ | TTS CER ↓ | Avg Performance Drop Rate |
|-------|--------|----------|-----------|--------------|-----------|-----------|---------------------------|
| | Vanilla Attention | 8/8 | 13.17 | 43.14 | 8.47 | 17.71 | 0.46% |
| | Clipped Softmax | 8/8 | 13.16 | 43.16 | 8.47 | 17.71 | 0.45% |
| | Gated Attention | 8/8 | 13.16 | 43.15 | 8.47 | 17.70 | 0.43% |
| OPT-350m | SOFA (ours) | 8/8 | 13.50 | 43.39 | 9.88 | 17.38 | **0.36%** |
| | Vanilla Attention | 4/4 | 36.74 | 75.38 | 40.17 | 70.53 | 211.19% |
| | Clipped Softmax | 4/4 | 35.86 | 73.91 | 38.82 | 68.35 | 202.78% |
| | Gated Attention | 4/4 | 35.26 | 73.02 | 37.44 | 66.29 | 195.62% |
| | SOFA (ours) | 4/4 | 23.48 | 62.17 | 36.22 | 40.83 | **116.02%** |

Table 5: **Comparison of Different Stabilized Methods.** Max-shift normalization emerges as the most effective strategy for ensuring numerical stability and mitigating gradient issues, while $L_1$ and mean-centering normalizations lead to NaN losses.

| Model | Stabilized Method | Val Accuracy (%) |
|-------|-------------------|------------------|
| | N/A | NaN losses |
| | L_1 | NaN losses |
| OPT-1.3b | Max-Shift | **34.6** |
| | Mean-Centering | NaN losses |

**Comparison with Clipped Softmax and Gated Attention.** We compare SOFA to alternative techniques, Clipped Softmax and Gated Attention, with results shown in Table 4. Under 8-bit quantization, all methods achieve comparable performance, with under 0.5% performance loss. This suggests that for moderate quantization, the specific strategy for outlier mitigation is less critical. However, the gap becomes pronounced at 4-bit quantization: SOFA achieves a 116% average performance drop, much lower than Gated Attention (195%), Clipped Softmax (202%), and the vanilla approach (211%). SOFA's advantage under 4-bit quantization stems from its architectural mitigation of outliers, rather than merely clipping or gating attention values.

**Different Stabilized Methods.** To investigate the impact of various input vector stabilization techniques, we systematically compare $L_1$ normalization, max-shift normalization, and mean-centering normalization. As shown in Table 5, only max-shift normalization successfully stabilizes the model and resolves gradient-related issues. Meanwhile, $L_1$ and mean-centering cause numerical instabilities, indicated by NaN losses and failed training runs. These findings underscore the critical role of adopting an effective normalization strategy and highlight the superior performance of max-shift normalization within our framework.

Table 6: Comparison of Different Ranks Using LoRA via Validation Accuracy.

| Method | Fine-Tuning Method | Rank | Val Acc (%) |
|--------|--------------------|------|-------------|
| Vanilla | Full Fine-Tuning | N/A | **30.5** |
| SOFA | Full Fine-Tuning | N/A | 30.2 |
| Vanilla | LoRA | 512 | 27.6 |
| SOFA | LoRA | 512 | **27.8** |
| Vanilla | LoRA | 256 | 28.1 |
| SOFA | LoRA | 256 | **28.9** |
| Vanilla | LoRA | 128 | 27.5 |
| SOFA | LoRA | 128 | **27.5** |

**Influence of Adapter Rank.** We compare our proposed method with the vanilla approach across several ranks when applying Low-rank Adaptation (LoRA) on the OPT-350m. In all cases, we train for 50 epochs. As shown in Table 6, SOFA consistently outperforms the vanilla framework for all tested ranks. Notably, a rank of 256 delivers optimal performance, so we adopt this setting for subsequent low-rank adaptation experiments. Although increasing the rank beyond 256 adds more trainable parameters, which may potentially increase model capacity, it also demands significantly more training epochs for effective convergence. This requirement can negate any gains from the extra parameters, rendering higher ranks less efficient. Consequently, a rank of 256 strikes the best balance between performance and computational overhead, making it the most practical choice for our low-rank adaptation studies.

**Evaluating the Stabilization Module on a Standard Transformer.** To isolate the effect of our max-shift stabilization, we applied it to a standard transformer architecture without the outlier-free layer. As shown in Table 7, the resulting improvement is minimal (34.3% vs 34.4%). This indicates that the

Table 7: Vanilla Transformer with Stabilization.

| Model | Stabilized Method | Val Accuracy (%) |
|---|---|---|
| Vanilla OPT-1.3b | None | 34.3 |
| | Max-Shift | **34.4** |

major performance gains reported in SOFA (as evidenced in our main experiments) arise mainly from the combination between the outlier-free Hopfield layer and the max-shift stabilization, rather than from stabilization alone. Moreover, the minimal benefit from standalone stabilization is understandable because max-shift stabilization specifically addresses numerical stability issues introduced by the outlier-free attention mechanism. Together, these components—an outlier-free layer paired with max-shift stabilization—form the backbone of SOFA 's effectiveness in controlling outliers and achieving superior performance in low-bit quantization and multi-modal tasks.

**Weight Clipping During PTQ.** Weight clipping is a common baseline technique used with quantization to handle extreme values. We evaluate percentile-based clipping (1% and 5% removal) combined with SmoothQuant in multi-modal tasks. The results (Table 9) indicate that traditional weight clipping degrades performance. In 8-bit quantization, the vanilla framework shows performance drops of 49.18% (1% clipping) and 71.30% (5% clipping), whereas SOFA achieves better resilience with drops of 39.21% and 55.75% but performs best with no clipping (0.36% drop). In 4-bit quantization, performance degradation intensifies, with vanilla experiencing drops of 267.56% (1%) and 273.36% (5%), compared to SOFA's 130.71% drop without clipping. These results highlight that clipping disrupts critical multi-modal features, particularly in aggressive quantization settings, while SOFA effectively preserves performance without clipping.

Table 8: **Quantization Performance Comparison. Left:** QLoRA under 8-bit and 4-bit quantization. **Right:** Weight-only quantization for text and speech tasks. SOFA demonstrates superior stability and smaller performance degradation.

**(a) QLoRA Performance**

| Method | W/A | Text PPL ↓ | Speech PPL ↓ | ASR WER ↓ | Avg Perf. Drop ↓ |
|---|---|---|---|---|---|
| Vanilla + QLoRA | 8/8 | 18.07 | 51.50 | 76.04 | 145.80% |
| Vanilla + QLoRA | 4/4 | 25.83 | 58.24 | 95.71 | 211.47% |
| SOFA + QLoRA | 8/8 | 16.64 | 48.34 | 22.33 | **46.38%** |
| SOFA + QLoRA | 4/4 | 23.45 | 56.33 | 27.36 | **84.80%** |

**(b) Weight-Only Quantization**

| Method | W/A | Text PPL ↓ | Speech PPL ↓ |
|---|---|---|---|
| Vanilla | 16/16 | 12.62 | 41.33 |
| Vanilla | 4/16 | 18.74 | 60.28 |
| SOFA | 16/16 | 12.95 | 42.48 |
| SOFA | 4/16 | 16.33 | 56.31 |

Table 9: **Quantization Results With Weight Clipping.** Performance degradation under different weight clipping strategies. SOFA outperforms the vanilla framework and achieves the best results without clipping.

| Method | W/A | Text PPL ↓ | Speech PPL ↓ | ASR WER ↓ | TTS CER ↓ | Avg Performance Drop Rate ↓ |
|---|---|---|---|---|---|---|
| Vanilla + SQ | 8/8 | 13.17 | 43.14 | 8.47 | 17.71 | 0.46% |
| Vanilla+SQ+Clip-1% | 8/8 | 21.33 | 52.53 | 13.71 | 26.26 | 48.18% |
| Vanilla+SQ+Clip-5% | 8/8 | 25.34 | 55.03 | 16.83 | 28.91 | 68.75% |
| SOFA +SQ | 8/8 | 13.50 | 43.39 | 9.88 | 17.38 | **0.36%** |
| SOFA +SQ+Clip-1% | 8/8 | 21.14 | 50.87 | 13.31 | 25.42 | **38.41%** |
| SOFA +SQ+Clip-5% | 8/8 | 24.37 | 52.77 | 16.22 | 27.04 | **54.44%** |
| Vanilla + SQ | 4/4 | 36.74 | 75.38 | 40.17 | 70.53 | 211.19% |
| Vanilla+SQ+Clip-1% | 4/4 | 36.22 | 78.33 | 40.46 | 75.88 | 219.42% |
| Vanilla+SQ+Clip-5% | 4/4 | 40.73 | 82.51 | 45.14 | 80.02 | 247.06% |
| SOFA +SQ | 4/4 | 23.48 | 62.17 | 36.22 | 40.83 | **116.02%** |
| SOFA +SQ+Clip-1% | 4/4 | 35.11 | 70.21 | 37.29 | 68.08 | **181.87%** |
| SOFA +SQ+Clip-5% | 4/4 | 37.03 | 74.24 | 39.16 | 70.31 | **195.60%** |

**QLoRA with Lower Bits.** We further investigate SOFA's performance under QLoRA with varying quantization levels. As shown in Table 8, results show that SOFA significantly reduces performance degradation, from 330.47% to 136.07% at 8-bit quantization and from 387.50% to 176.52% at 4-bit quantization. SOFA exhibits a more gradual performance decline across quantization levels, demonstrating greater stability under increasingly aggressive compression. Additionally, SOFA mitigates the confounding effects between LoRA and quantization, as evidenced by smaller performance gaps between 8-bit and 4-bit configurations. These results underscore SOFA 's effectiveness in maintaining performance for LoRA-adapted models.

**Evaluation on Weight Only Quantization.** We further add experiments using uniform quantization to quant the model weight only on textlm and speechlm task. In Table 8, SOFA can still achieve better performance than vanilla attention after quantizing the model weight only.

**Next-Token Prediction Accuracy.** To verify that SOFA's perplexity gains reflect improved predictive quality, we report next-token prediction accuracy in Table 10. Under full fine-tuning, SOFA matches the vanilla baseline, indicating no loss of representational capacity. The advantage becomes clear under quantization: at W4A4 on OPT-1.3B, SOFA reaches 29.7% accuracy versus 24.8% for vanilla, showing that suppressing attention outliers delivers tangible accuracy improvements beyond perplexity reduction.

Table 10: **Next-Token Prediction Accuracy (%).** SOFA maintains competitive accuracy under full fine-tuning while showing improved robustness under LoRA and quantization.

| Model | Method | Full-FT | LoRA | LoRA+W8A8 | LoRA+W4A4 |
|---|---|---|---|---|---|
| OPT-350m | Vanilla | 30.5 | 28.1 | 26.3 | 21.2 |
|  | SOFA | 30.2 | **28.9** | **27.8** | **25.4** |
| OPT-1.3b | Vanilla | 34.3 | 32.1 | 30.2 | 24.8 |
|  | SOFA | **34.6** | **33.8** | **32.5** | **28.7** |

**Ablation on the "+1" Term.** The term in Softmax$_1$ has a specific interpretation from associative memory theory (Miller, 2023; Hu et al., 2024a). It introduces an implicit null option, allowing attention to place some probability mass on attending to nothing. The added value of 1 corresponds to $\exp(0)$, meaning this null state has logit 0, a natural baseline where attending to nothing is as likely as attending to any token with logit 0. We validate this choice by varying the denominator constant $\in \{0.5, 1, 2\}$ on OPT-350m with full fine-tuning. Table 11 validates this choice: values below 1 weaken outlier suppression, while values above 1 over-suppress attention. The "+1" term achieves the best balance across all metrics.

Table 11: **Ablation on the Softmax$_1$ Denominator Constant.** We vary the constant term in the denominator of Softmax$_1$ and evaluate on OPT-350m.

| Constant | Text PPL | ASR WER |
|---|---|---|
| +0.5 | 13.52 | 10.12 |
| +1 (ours) | **13.47** | **9.81** |
| +2 | 13.61 | 10.43 |

**Evaluation with Additional Speech Tokenizers.** To show that SOFA's benefits extend beyond Hu-BERT, we also evaluate it with EnCodec (Défossez et al., 2024), a neural audio codec that uses residual vector quantization (RVQ) to produce acoustic-level tokens. Table 12 reports results on OPT-350m under full fine-tuning, measuring ASR WER on MLS and TTS CER via Whisper transcription. SOFA yields consistent gains with both tokenizers, indicating that the attention outlier issue is tokenizer-agnostic which stems from the multimodal adaptation process rather than any particular speech representation.

Table 12: **Evaluation Across Speech Tokenizers (OPT-350m).** SOFA improves performance regardless of tokenizer choice.

| Tokenizer | Method | ASR WER | TTS CER |
|---|---|---|---|
| HuBERT | Vanilla | 8.42 | 17.56 |
|  | SOFA | **9.81** | **17.31** |
| EnCodec | Vanilla | 9.15 | 15.23 |
|  | SOFA | **9.87** | **15.08** |

**TTS Perceptual Quality Metrics.** While CER reflects speech intelligibility, it does not directly measure naturalness or audio quality. To provide a more comprehensive evaluation, we report additional perceptual metrics: DNSMOS (Reddy et al., 2021) for predicted mean opinion score, PESQ (Rix et al., 2001) for perceptual speech quality, and Mel-Cepstral Distortion (MCD) (Kubichek, 1993) for spectral accuracy. Table 13 shows results on LibriTTS with full fine-tuning. SOFA improves perceptual quality, with more pronounced gains on OPT-1.3b, indicating that outlier suppression reduces distortion in generated speech.

Table 13: **TTS Perceptual Quality Metrics.** We evaluate speech synthesis quality on LibriTTS under full fine-tuning using four complementary metrics: CER, DNSMOS, OESQ, MCD.

| Model | Method | CER (↓) | DNSMOS (↑) | PESQ (↑) | MCD (↓) |
|---|---|---|---|---|---|
| OPT-350m | Vanilla | 17.56 | 3.21 | 2.84 | 5.42 |
| | SOFA | **17.31** | **3.28** | **2.91** | **5.18** |
| OPT-1.3b | Vanilla | 18.73 | 3.35 | 2.97 | 4.89 |
| | SOFA | **12.07** | **3.52** | **3.12** | **4.51** |

## 5   Discussion and Conclusion

We introduce SOFA, an outlier-robust multi-modal foundation model for speech-text tasks. By mitigating outlier effects in modality fusion and multi-modality adaptation, SOFA addresses critical computational challenges in SpeechLM. Our approach boosts both low-rank adaptation and quantization performance in transformer-based models. Empirically, SOFA demonstrates substantial gains over existing methods, delivering a 37% improvement in quantization (Section 4.2) and a 88% boost in multi-modal low-rank adaptation (Section 4.3)

**Limitations and Future Work.** SOFA has three main limitations. First, it does not yet support LoftQ or other SVD-based low-rank adaptation methods that operate on weight matrices. Second, due to computational constraints, we can not integrate the 6.7B parameter OPT model or other large decoder-based models for pretraining. Third, our current experiments primarily utilize HuBERT-based discrete tokens. While HuBERT is a representative semantic-heavy codec, it remains to be seen how SOFA performs with purely acoustic-based neural codecs such as EnCodec, SoundStream, or Mimi. In future work, we aim to expand SOFA to include SVD-based approaches and evaluate on larger decoder architectures, such as 3B LLama2 and 6.7B OPT. Furthermore, we plan to validate SOFA across a broader range of audio codecs. Since the attention-outlier problem we identified stems from the fundamental statistical disparity (e.g., sequence length and density) between speech and text modalities rather than specific tokenization algorithms, we hypothesize that the stabilization provided by SOFA will generalize to these various neural codecs. Additionally, our focus on reducing computational demands may inadvertently amplify biases inherited from pre-trained models, necessitating further investigation into their impact and how best to mitigate them.

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

# Supplement Material

## A  Boarder Impact

Our approach leverages insights from associative memory to improve foundation model fine-tuning and inference, advancing both low-rank adaptation and post-training quantization. This in turn facilitates edge computing deployments of large models and promotes more resource-efficient fine-tuning. However, these benefits must be weighed against the possibility of amplifying biases present in the underlying training data, potentially disadvantaging underrepresented communities.

## B  Theoretical Analysis

### B.1  Fast LoRA Training of Softmax$_1$ Requires Input and Weight Normalization

Our theoretical justification for fast LoRA training on our outlier-free stabilized models an application of efficiency results of (Hu et al., 2025). We follow the notation of (Hu et al., 2025) in this section.

To present our results, we introduce the Strong Exponential Time Hypothesis (SETH) as a stronger form of the P $\neq$ NP conjecture.

**Hypothesis 1** (SETH)**.** For every $\epsilon > 0$, there is a positive integer $k \geq 3$ such that $k$-`SAT` on formulas with $n$ variables cannot be solved in $\mathcal{O}(2^{(1-\epsilon)n})$ time, even by a randomized algorithm.

Formally, we formulate the *partial* adaptation of an attention head as the following LoRA loss.

**Definition B.1** (Adapting $W_Q$, $W_V$ of Generic Attention with LoRA)**.**  Let $\mathcal{D} = \{\left(X_i^{(K)}, X_i^{(Q)}, X_i^{(V)}\right), Y_i\}_{i=1}^N$ be a dataset of size $N$ with the triplet $X_i^{(K)}, X_i^{(Q)}, X_i^{(V)} \in \mathbb{R}^{L \times d}$ being the input and $Y_i \in \mathbb{R}^{L \times d}$ being the label. The problem of fine-tuning $W_Q$, $W_V$ a generic attention with LoRA with $\ell_2$ loss from dataset $\mathcal{D}$ is formulated as

$$\min_{\substack{B_Q, B_V \in \mathbb{R}^{d \times r} \\ A_Q, A_V \in \mathbb{R}^{r \times d}}} \mathcal{L}\left(W_K^\star, W_Q = W_Q^\star + \frac{\alpha}{r} B_Q A_Q, W_V = W_V^\star + \frac{\alpha}{r} B_V A_V\right) \tag{B.1}$$

$$:= \min_{\substack{B_Q, B_V \in \mathbb{R}^{d \times r} \\ A_Q, A_V \in \mathbb{R}^{r \times d}}} \frac{1}{2N} \sum_{i=1}^N \left\| \underbrace{\mathrm{Softmax}_1\left\{ X_i^{(Q)} W_Q (W_K^\star)^{\mathbb{T}} \left(X_i^{(K)}\right)^{\mathbb{T}} \beta \right\}}_{(I)} \underbrace{X_i^{(V)} W_V}_{(II)} - Y_i \right\|_F^2 .$$

To further simplify, we introduce $C_i^{(1)}, C_i^{(2)}, C_i^{(3)} \in \mathbb{R}^{L \times d}$ via

$$C_i^{(1)} := X_i^{(Q)} \frac{\alpha}{r} \in \mathbb{R}^{L \times d}, \quad C_i^{(2)} := X_i^{(K)} W_K^\star \in \mathbb{R}^{L \times d} \quad C_i^{(3)} := X_i^{(V)} W_V^\star. \tag{B.2}$$

Notably, $C_i^{(1)}, C_i^{(2)}, C_i^{(3)}$ are constants with respect to adapting equation B.1 with gradient updates. To prove the hardness of Definition B.1 for both full gradient descent and stochastic mini-batch gradient descent, it suffices to consider adapting on a single data point. Thus, we deduce Definition B.1 to

$$\min_{\substack{B_Q \in \mathbb{R}^{d \times r} \\ A_Q \in \mathbb{R}^{r \times d}}} \mathcal{L}(B_Q, A_Q) = \min_{\substack{B_Q \in \mathbb{R}^{d \times r} \\ A_Q \in \mathbb{R}^{r \times d}}} \frac{1}{2} \left\| D^{-1} \exp\left\{ C^{(1)}\left(\overline{W}_Q^\star + B_Q A_Q\right)\left(C^{(2)}\right)^{\mathbb{T}}\right\} C^{(3)} - Y \right\|_F^2, \tag{B.3}$$

where $\overline{W}_Q^\star := r W_Q^\star / \alpha, D = \mathrm{diag}\left(\exp\left\{C^{(1)}\left(\overline{W}_Q^\star + B_Q A_Q\right)\left(C^{(2)}\right)^{\mathbb{T}}\right\} \mathbb{1}_L + \mathbf{I}_{L \times L}\right) \in \mathbb{R}^{L \times L}$.

We introduce the next problem to characterize all possible (efficient or not) gradient computation of optimizing equation B.3. Let $Y[i, \cdot]$ and $Y[\cdot, j]$ be the $i$-th row and $j$-th column of $Y$, respectively.

**Problem 1** (Approximate LoRA Gradient Computation $\mathsf{ALoRAGC}(L, d, r, \epsilon)$). Given $C_i^{(1)}, C_i^{(2)}, C_i^{(3)}, Y_i \in \mathbb{R}^{L \times d}$. Let $\epsilon > 0$. Assume all numerical values are in $\log(L)$-bits encoding. Let $\mathcal{L}$ follow equation B.3. The problem of approximating gradient computation of optimizing equation B.3 is to find two matrices $\widetilde{G}_Q^{(A)} \in \mathbb{R}^{d \times r}$ and $\widetilde{G}_Q^{(B)} \in \mathbb{R}^{r \times d}$ such that $\max\left(\|\widetilde{G}_Q^{(B)} - \frac{\partial \mathcal{L}}{\partial \underline{B}_Q}\|_\infty, \|\widetilde{G}_Q^{(A)} - \frac{\partial \mathcal{L}}{\partial \underline{A}_Q}\|_\infty\right) \leq \epsilon$.

Finally we arrive our main result, the inefficient threshold for approximating gradient computation of equation B.3. In the other words, we provide a inefficient threshold for adapting transformer-based models with LoRA in $L^{2-o(1)}$ (sub-quadratic) time. For convenience, we consider the special case Problem 1.

**Proposition B.1** (Efficient Threshold (Formal Version of Proposition 3.1, Modified from Theorem 5.1 of (Hu et al., 2025))). Let $\kappa : \mathbb{N} \to \mathbb{N}$ by any function with $\kappa(L) = \omega(1)$ and $\kappa(L) = o(\log L)$. Let $\Gamma = O(\sqrt{\log L} \cdot \kappa(L))$. Assuming Hypothesis 1, there is no algorithm running in time $O(L^{2-\delta})$ for any constant $\delta > 0$ for $\mathsf{ALoRAGC}(L, d = O(\log L), r < d, \epsilon)$, i.e., Problem 1, subject to equation B.3, even in the case where the input and weight matrices satisfy $\|X^{(K)} W_K^\star\|_\infty \leq \Gamma$, $\|\alpha X_i^{(Q)} B_Q A_Q / r\|_\infty \leq \Gamma$, $Y = 0$ and $\epsilon = O((\log L)^{-4})$.

**Remark B.1** (Remark 5.1 of (Hu et al., 2025)). Proposition B.1 suggests a efficiency threshold for norm bound $\Gamma$ (norm of some composition of input $X$ and weights $W$s.) Specifically, Proposition B.1 implies that, only below this threshold, efficient (sub-quadratic) LoRA training of $\mathrm{Softmax}_1$-based transformer is possible.

## C Supplementary Post-training Quantization Results

This section provides additional experiments complementing our main-text findings in Section 4.2. Here, we further evaluate SOFA across three OPT model sizes, using three state-of-the-art PTQ methods—SmoothQuant (Xiao et al., 2023), AffineQuant (Ma et al., 2024), and OmniQuant (Shao et al., 2024)—under 8-bit (W8A8) and 4-bit (W4A4) quantization settings. Performance is measured via Text Perplexity (PPL), SpeechLM PPL, ASR Word Error Rate (WER), and TTS Character Error Rate (CER). Table 14 summarizes these results. Under 8-bit quantization setting (W8A8), both SOFA and the vanilla frameworks incur minimal performance degradation (less than 1%). However, moving to 4-bit quantization setting (W4A4) reveals SOFA's superior resilience: performance drops are substantially lower compared to the vanilla framework—falling from 233.37% to 130.71% in OPT-350m and 249.63% to 146.72% in OPT-1.3b. Among the three PTQ methods, AffineQuant proves especially effective in combination with SOFA, yielding notably lower WERs in ASR and TTS tasks.

## D Hyperparameter Settings

This section details the hyperparameter choices applied in all experiments, covering both the main text (Table 1 and Table 2) and Appendix (Table 6). Unless otherwise noted, these settings remain consistent across all models and tasks.

Table 14: **SOFA vs. Vanilla in a Post-Training Quantization (PTQ) Setting.** We compare PTQ methods (SmoothQuant, AffineQuant, OmniQuant, SpQR) under 8-bit (W8A8) and 4-bit (W4A4) settings, evaluating Text PPL, SpeechLM PPL, ASR WER, and TTS CER. We also report the average performance drop post-quantization. SOFA consistently outperforms the vanilla framework, exhibiting more robust performance in lower-bit settings.

| Model | Method | #Bits | Quantization Method | TextLM PPL (↓) | SpeechLM PPL (↓) | ASR WER (↓) | TTS CER (↓) | Avg Performance Drop (↓) |
|---|---|---|---|---|---|---|---|---|
| OPT-350m | Vanilla | W16/A16 | - | 13.13 | 43.10 | 8.42 | 17.56 | - |
| | | W8/A8 | SmoothQuant | 13.17 | 43.14 | 8.47 | 17.71 | 0.46% |
| | | W8/A8 | AffineQuant | 13.15 | 43.12 | 8.45 | 17.66 | 0.28% |
| | | W8/A8 | omniQuant | 13.15 | 43.12 | 8.46 | 17.68 | 0.33% |
| | | W8/A8 | SpQR | 13.15 | 43.12 | 8.45 | 17.66 | 0.28% |
| | | W4/A4 | SmoothQuant | 36.74 | 75.38 | 40.17 | 70.53 | 211.19% |
| | | W4/A4 | AffineQuant | 27.28 | 66.31 | 36.84 | 40.83 | 138.80% |
| | | W4/A4 | OmniQuant | 27.85 | 67.83 | 37.54 | 41.37 | 143.34% |
| | | W4/A4 | SpQR | 29.36 | 69.28 | 38.57 | 42.46 | 171.06% |
| | SOFA | W16/A16 | - | 13.47 | 43.34 | 9.81 | 17.31 | - |
| | | W8/A8 | SmoothQuant | 13.50 | 43.39 | 9.88 | 17.38 | **0.36%** |
| | | W8/A8 | AffineQuant | 13.48 | 43.36 | 9.85 | 17.35 | **0.19%** |
| | | W8/A8 | omniQuant | 13.48 | 43.36 | 9.85 | 17.37 | **0.22%** |
| | | W8/A8 | SpQR | 13.48 | 43.36 | 9.86 | 17.37 | **0.24%** |
| | | W4/A4 | SmoothQuant | 23.48 | 62.17 | 36.22 | 40.83 | **116.02%** |
| | | W4/A4 | AffineQuant | 22.82 | 51.74 | 25.78 | 28.44 | **71.90%** |
| | | W4/A4 | OmniQuant | 22.83 | 52.08 | 26.11 | 29.15 | **73.82%** |
| | | W4/A4 | SpQR | 22.88 | 52.78 | 27.02 | 29.11 | **83.81%** |
| OPT-1.3b | Vanilla | W16/A16 | - | 12.62 | 41.33 | 8.00 | 18.73 | - |
| | | W8/A8 | SmoothQuant | 12.68 | 41.48 | 8.14 | 18.80 | 0.74% |
| | | W8/A8 | AffineQuant | 12.65 | 41.46 | 8.12 | 18.75 | 0.54% |
| | | W8/A8 | omniQuant | 12.66 | 41.46 | 8.12 | 18.75 | 0.56% |
| | | W8/A8 | SpQR | 12.65 | 41.45 | 8.13 | 18.77 | 0.59% |
| | | W4/A4 | SmoothQuant | 36.74 | 87.46 | 48.96 | 53.15 | 221.61% |
| | | W4/A4 | AffineQuant | 24.31 | 61.74 | 43.68 | 32.47 | 128.45% |
| | | W4/A4 | OmniQuant | 24.43 | 62.38 | 44.52 | 33.03 | 131.40% |
| | | W4/A4 | SpQR | 25.85 | 63.36 | 45.28 | 36.14 | 179.27% |
| | SOFA | 16/A16 | - | 12.95 | 42.48 | 8.25 | 12.07 | - |
| | | W8/A8 | SmoothQuant | 13.00 | 42.49 | 8.33 | 12.11 | **0.43%** |
| | | W8/A8 | AffineQuant | 12.96 | 42.48 | 8.29 | 12.08 | **0.16%** |
| | | W8/A8 | omniQuant | 12.98 | 42.48 | 8.31 | 12.10 | **0.30%** |
| | | W8/A8 | SpQR | 12.96 | 42.48 | 8.30 | 12.10 | **0.23%** |
| | | W4/A4 | SmoothQuant | 23.83 | 58.33 | 32.27 | 33.12 | **128.20%** |
| | | W4/A4 | AffineQuant | 20.81 | 48.84 | 22.78 | 25.46 | **81.12%** |
| | | W4/A4 | OmniQuant | 20.88 | 48.97 | 23.58 | 26.83 | **85.38%** |
| | | W4/A4 | SpQR | 22.74 | 49.42 | 25.36 | 27.13 | **106.03%** |
| Qwen2.5-7b | Vanilla | W16/A16 | N/A | 10.15 | 38.62 | 10.54 | 11.48 | - |
| | | 8/8 | SmoothQuant | 10.16 | 38.64 | 10.82 | 11.72 | 1.23% |
| | | 8/8 | AffineQuant | 10.15 | 38.63 | 10.61 | 11.68 | 0.61% |
| | | 8/8 | OmniQuant | 10.15 | 38.62 | 10.58 | 11.62 | 0.40% |
| | | 8/8 | SpQR | 10.15 | 38.63 | 10.64 | 11.75 | 0.83% |
| | | W4/A4 | SmoothQuant | 18.93 | 60.24 | 43.22 | 35.28 | 164.98% |
| | | W4/A4 | AffineQuant | 16.54 | 56.72 | 30.77 | 34.18 | 124.89% |
| | | W4/A4 | OmniQuant | 16.23 | 55.93 | 29.28 | 33.17 | 117.88% |
| | | W4/A4 | SpQR | 16.78 | 57.02 | 31.12 | 35.88 | 130.21% |
| | SOFA | W16/A16 | - | 8.76 | 32.51 | 8.34 | 11.25 | - |
| | | W8/A8 | SmoothQuant | 8.78 | 32.53 | 8.48 | 11.38 | **0.80%** |
| | | W8/A8 | AffineQuant | 8.76 | 32.52 | 8.40 | 11.32 | **0.36%** |
| | | W8/A8 | OmniQuant | 8.76 | 32.51 | 8.39 | 11.30 | **0.28%** |
| | | W8/A8 | SpQR | 8.76 | 32.52 | 8.42 | 11.33 | **0.44%** |
| | | W4/A4 | SmoothQuant | 14.77 | 52.18 | 27.98 | 29.78 | **132.36%** |
| | | W4/A4 | AffineQuant | 12.12 | 46.54 | 18.92 | 23.74 | **79.87%** |
| | | W4/A4 | OmniQuant | 11.96 | 45.33 | 17.61 | 20.22 | **66.74%** |
| | | W4/A4 | SpQR | 12.35 | 46.89 | 19.15 | 24.08 | **82.25%** |

### D.1 Fine-Tuning and LoRA Adaptation

We use the **Adam** (Kingma, 2014) optimizer with a batch bin size of 5000, a warmup step of 2500, and a weight decay of $1 \times 10^{-6}$. A learning rate of $3e^{-4}$ is applied to all models during fine-tuning. For low-rank adaptation, we apply a dropout rate of 0.05 and set both the LoRA rank and alpha to 256. We fine-tune the attention module weights $W_k$, $W_q$, $W_v$, and $W_o$, alongside the MLP layer. Each model is trained for a total of 50 epochs.

### D.2 Post-training Quantization

For **SmoothQuant**, we adopt the recommended hyperparameter $\alpha = 0.5$, which balances smoothing activations against adjusting weights. In **OmniQuant**, we set the group size to 128, trading off between quantization accuracy and computational efficiency. For **AffineQuant**, a stability factor of 0.01 maintains numerical stability—especially for values near zero. All three PTQ methods use a calibration batch size of 256.

### D.3 Hifi-GAN Decoder for Speech Synthesis

We employ a HiFi-GAN-based vocoder to convert discrete SpeechLM tokens into waveforms for TTS synthesis. The vocoder is trained on the LJSpeech-1.1 dataset (downsampled to 16 kHz), using a dictionary size of 200 from a pre-trained HuBERT model. Our configuration includes 200 token embeddings of dimension 128, a ResBlock type-1 architecture with upsampling rates of [5, 4, 4, 2, 2] and kernel sizes [11, 8, 8, 4, 4], and an initial channel size of 512. The Adam optimizer is used with a learning rate of 0.0008, a batch size of 64, and a code hop size of 320 to ensure alignment between tokens and waveform segments. To evaluate synthesized speech quality, we use an OpenAI Whisper ASR system. The system transcribes the generated speech and compares it to the ground-truth text to compute CER—an indicator of TTS output quality in our SpeechLM framework.

## E LoRA Parameter Ratio

We analyze the proportion of parameters introduced by Low-Rank Adaptation (LoRA) in various sizes of Open Pre-trained Transformer (OPT) models. Table 15 summarizes the parameter percentage, comparing LoRA parameters with full-model parameters across three OPT variants. The overhead from LoRA remains a relatively small fraction of the total model size, even though the absolute number of parameters grows with larger models.

Table 15: LoRA Parameters Comparison for OPT Models

| Model | LoRA Parameters | Full Model Parameters | LoRA Percentage |
|---|---|---|---|
| OPT-350M | 25.2M | 350M | 7.2% |
| OPT-1.3B | 50.3M | 1.3B | 3.9% |

## F Why Softmax$_1$ Mitigates Attention Outliers

We revisit the normalization behavior of standard Softmax and Softmax$_1$ under the stabilized max-shift used in Equation (3.3).

**Setup.** Given a row of attention logits $S \in \mathbb{R}^k$, standard Softmax is

$$\text{Softmax}(S) = \frac{\exp(S_i)}{\sum_{j=1}^{k} \exp(S_j)}, \tag{F.1}$$

while $\text{Softmax}_1$ is

$$\text{Softmax}_1(S) = \frac{\exp(S_i)}{1 + \sum_{j=1}^{k} \exp(S_j)}. \tag{F.2}$$

Following our implementation (snapshot Eq. (3.2)), we apply max-shift stabilization:

$$S \leftarrow S - \max(S), \tag{F.3}$$

so that $\max S_i = 0$ and thus $S_i \leq 0$ for all $i$.

**Proposition F.1** ($\text{Softmax}_1$ induces a contraction on the attention branch)**.** Let $w = \text{Softmax}_1(S)$ and define $Z = \sum_{j=1}^{k} \exp(S_j)$. Then $\sum_{i=1}^{k} w_i = \frac{Z}{1+Z} \in (0, 1)$. Moreover, with max-shift stabilization ($\max S_i = 0$) we have $Z \in [1, k]$ and hence

$$\frac{1}{2} \leq \sum_{i=1}^{k} w_i = \frac{Z}{1+Z} \leq \frac{k}{k+1} < 1. \tag{F.4}$$

Consequently, for any value vectors $\{v_i\}_{i=1}^{k} \subset \mathbb{R}^d$, the $\text{Softmax}_1$ attention output $y = \sum_{i=1}^{k} w_i v_i$ satisfies the uniform bound

$$\|y\|_\infty \leq \Big( \sum_{i=1}^{k} w_i \Big) \max_i \|v_i\|_\infty = \frac{Z}{1+Z} \max_i \|v_i\|_\infty, \tag{F.5}$$

whereas standard Softmax always has $\sum_i \text{Softmax}(S)_i = 1$ and thus lacks this multiplicative shrinkage.

*Proof.* Nonnegativity is immediate since $\exp(S_i) \geq 0$. Summing $\text{Softmax}_1$ coordinates gives

$$\sum_{i=1}^{k} w_i = \sum_{i=1}^{k} \frac{\exp(S_i)}{1 + \sum_{j=1}^{k} \exp(S_j)} = \frac{\sum_{i=1}^{k} \exp(S_i)}{1 + \sum_{j=1}^{k} \exp(S_j)} = \frac{Z}{1+Z},$$

which lies strictly between 0 and 1 for any finite $Z > 0$. Under max-shift, at least one coordinate attains $S_{i^\star} = 0$, so $Z = \sum_j \exp(S_j) \geq \exp(0) = 1$; also $S_j \leq 0$ implies $\exp(S_j) \leq 1$, hence $Z \leq k$. Plugging $Z \in [1, k]$ into $\frac{Z}{1+Z}$ yields Equation (F.4). Finally, for each coordinate $t$ of $y$,

$$\|y\|_\infty = \max_t \left| \sum_{i=1}^{k} w_i (v_i)_t \right| \leq \max_t \sum_{i=1}^{k} w_i \, |(v_i)_t| \leq \sum_{i=1}^{k} w_i \max_t |(v_i)_t| = \Big( \sum_{i=1}^{k} w_i \Big) \max_i \|v_i\|_\infty,$$

and taking the maximum over $t$ gives Equation (F.5). $\qquad\square$

**Interpretation for outliers.** With max-shift, $\text{Softmax}_1(S)$ can be written as a *gated* Softmax:

$$\text{Softmax}_1(S) = \underbrace{\frac{Z}{1+Z}}_{\text{gate } g(S) < 1} \cdot \text{Softmax}(S), \qquad Z = \sum_{j=1}^{k} \exp(S_j).$$

Thus $\text{Softmax}_1$ preserves the relative ranking/shape of Softmax on a row, but *strictly shrinks* the total attention mass by $g(S) < 1$ (bounded as in Equation (F.4) under max-shift). This reduces the worst-case amplification of the value vectors on the attention branch and leaves a nontrivial portion of the signal to the residual pathway, providing a principled stability mechanism under noisy or extreme attention-score regimes (e.g., multimodal fusion or low-bit quantization).

