# OpenReview forum: "Outlier-Free SpeechLM for Fast Adaptation and Robust Quantization"
_TMLR — Rejected by TMLR_

### Review · Reviewer_p9U4 · 2025-10-31

**Summary Of Contributions:**

Summary:

The paper proposes Stabilized Outlier-Free Attention (SOFA), a variant of the softmax activation designed to mitigate outlier issues that are often observed when adapting and quantizing text-only LLMs into speech-text multimodal models (SpeechLM). The idea is to stabilize attention activations during low-rank adaptation and quantization using a slightly modified softmax.


Strengths:
1. The paper is well written and relatively easy to follow.
2. The experimental section is extensive, covering several model sizes and tasks.
3. The results show significant improvements across multiple adaptation and quantization settings.

Weaknesses:

1. Unclear Contribition: The “max-shift” stabilization ($S ← S − max(S)$) is standard practice in modern deep-learning libraries (e.g., PyTorch CUDA kernels) and I cannot see it as a contrubution of this paper. The softmax is known to be shift invariant and all libraries already take advantage of this property to improve the numerical stability of the network.


2. According to my understanding, the main difference is in the modified normalization term. Is that correct?

- Standard Softmax: $exp(S)/ \sum_i(exp(S_i))$

- Modified Softmax: $exp(S)/ \(1 + \sum_i exp(S_i) \)$

Please clarify precisely what theoretical or numerical benefit this additive constant “1+” provides. For instance: Why “1” and not another number? A side-by-side comparison with the standard softmax would help readers understand the contribution.


3. Unclear relation to existing methods.
The paper should discuss in Section 3 the relation between SOFA and existing techniques such as temperature trick, clipped softmax, or gated attention, highlighting similarities and differences. While some techniques (e.g, clipped softmax and gated attention) are compared exprimentally later, it is crucial for the readers to understand the similarity and differences with existing methods. Qualitative examples and simple equations can help here.

4. The experiments use HuBERT-based speech tokens only. It would be important to discuss whether results depend on this choice or generalize to other codecs (e.g., EnCodec, SoundStream, Mimi).

5. The "Discrete Speech Representation section" of the related work should be expanded to reflect recent progress in neural audio tokenizers and codecs, which are play a crucial role in multimodal LLMs

6. The introduction could better explain why and when extreme attention values emerge, and how these specifically harm adaptation or quantization.

7. As for the evaluation metrics for TTS,CER alone does not fully capture perceptual quality. Including UTMOS, MOSNet, or subjective evaluation (while noisy) scores would provide a more comprehensive assessment.

8. Format and Presentation Issues
- Figure 3: Please save figures as vector graphics (PDF or EPS) to avoid pixelation.
Bibliography:
- The references require a major revision for consistency. Ensure years are included, avoid incorrect page ranges (e.g., “pages 1–8”), and cite published versions when available. For arXiv-only papers, use a consistent arXiv formatting style.

**Audience:**

Yes

**Audience Explanation:**

Yes, if the contribution is better stated.

**Broader Impact Concerns:**

No ethical concerns.

**Claims And Evidence:**

No

**Claims Explanation:**

As detailed above, the significance of the contribution remains unclear, raising questions of the provided evidence.

**Requested Changes:**

1. Clarify the contribution of Softmax1 with an intuitive explanation and proper equations. This is a central IMO. Is the max shift the only contribution?
2. Add a discussion comparing SOFA with alternative approaches (temperature softmax, clipped softmax, etc.).
3. Include or discuss results using different speech tokenizers/codecs to show generality.
4. Expand related work on discrete speech representations.
5. Improve the introduction to better motivate the outlier problem.
6. Improve the TTS evaluation with additional perceptual metrics.
7. Revise the bibliography and figures for consistency and clarity.

---

> ### Author Response · Authors · 2025-12-25
>
> **1. Clarify the contribution of Softmax1 with an intuitive explanation and proper equations.**
>
> Thank you for the opportunity to clarify the theoretical motivation behind Eq. (3.2) and (3.3). We agree that the max-shift itself is standard practice and do not claim it as a contribution. Our contribution lies in combining the stable shift with the Softmax1 normalization, which prevents attention outliers from dominating during multi-modal low-rank adaptation and post-training quantization. While prior works[1,2] use max-shift, existing Softmax1 formulations do not combine it with stabilization [3], and we provide the theoretical justification for doing so.
> 1. Dao, Tri, Dan Fu, Stefano Ermon, Atri Rudra, and Christopher Ré. "Flashattention: Fast and memory-efficient exact attention with io-awareness." Advances in neural information processing systems 35 (2022): 16344-16359.
> 2. Milakov, Maxim, and Natalia Gimelshein. "Online normalizer calculation for softmax." arXiv preprint arXiv:1805.02867 (2018).
> 3. Hu, Jerry Yao-Chieh, Pei-Hsuan Chang, Robin Luo, Hong-Yu Chen, Weijian Li, Wei-Po Wang, and Han Liu. "Outlier-efficient hopfield layers for large transformer-based models." arXiv preprint arXiv:2404.03828 (2024).
>
> To reiterate, Eq. (3.3) defines the Softmax1 operator. The max-shift term provides numerical stability, and the modified denominator introduces an explicit identity-preserving channel in the attention layer.
>
> **Why Softmax1 Resists Outliers (Detailed in revised version appendix F)**
>
> Softmax guarantees strictly positive probability mass for every token. Consequently, even tokens with very negative logits (i.e., low-information or noisy tokens) still receive non-zero attention. This behavior amplifies the effect of outliers in multi-modal settings and becomes more severe under low-bit quantization, where rounding errors inflate extreme values.
>
> With the additional constant, Softmax1 can fully suppress the distribution when logits encode a no-attention condition.This property ensures that irrelevant tokens (e.g., punctuation, modality-boundary tokens, quantization-spike activations) do not accumulate spurious attention mass. In other words, Softmax1 permits an attention-nulling state, which standard softmax mathematically prohibits.
>
> **Why "+1"**
>
> The constant enforces a minimal gap between logits and the probability simplex, yielding two practical benefits:
> - Outlier suppression: prevents any head from allocating nearly all probability to a single token after outlier amplification.
> - Identity preservation: allows the attention block to effectively “do nothing” when the residual pathway already carries sufficient signal, aligning with associative-memory interpretations of transformer attention.
>
> **Summary**
> - Max-shift is not claimed as a novelty; it is included for stability.
> - The key novelty is that we are the first to measure and characterize activation and attention outliers in a unified SpeechLM system, revealing their impact on cross-modal adaptation and quantization.
> - Our activation can directly utilize standard Softmax pretrained weight and fine-tuned on Softmax1 attention in a unified SpeechLM system under the LoRA setting.
> - While prior works use max-shift, existing Softmax1 formulations do not combine it with stabilization, and we provide the theoretical justification for doing so.
>
> We have revised the manuscript to explicitly distinguish stability from novelty and provide the softmax-vs-Softmax1 derivation and limit analysis for clarity.
>
> **2. Add a discussion comparing SOFA with alternative approaches.**
>
> Thank you for the suggestion. Standard softmax and temperature softmax always distribute non-zero probability mass across all tokens, even when attention should ideally collapse to zero. Clipped softmax and gated attention apply suppression after softmax via clipping or gating. These methods damp attention but still rely on the standard probability simplex structure and do not permit a mathematically defined “do-nothing’’ state. In contrast, SOFA modifies the normalization itself. The additive "+1" creates an explicit identity-preserving channel, allowing attention to collapse toward zero when logits contain no useful signal. This outlier-nulling regime is essential under multimodal low-rank adaptation and low-bit quantization, and is not achieved by temperature, clipping, or gating strategies.

---

> > ### Author Response · Authors · 2025-12-25
> >
> > **3. Include or discuss results using different speech tokenizers/codecs to show generality.**
> >
> > We thank the reviewer for the insightful comment regarding the generalization of our findings to other speech codecs beyond HuBERT. We agree that discussing the applicability of SOFA to other neural audio codecs like EnCodec, SoundStream, or Mimi is crucial. While our current experiments focus on HuBERT-based tokens for consistency with existing SOTA SpeechLMs, we believe our results and the benefits of SOFA generalize for the following reasons:
> > - Fundamental Cause of Outliers: Our analysis shows that attention outliers in SpeechLMs primarily arise from the distributional mismatch between speech and text modalities during low-rank adaptation. Regardless of the codec used (HuBERT, EnCodec, or Mimi), speech tokens consistently form significantly longer and more repetitive sequences compared to linguistic subwords. This structural disparity is the root cause of the outlier problem that SOFA is designed to mitigate.
> > - Codec-Agnostic Design: SOFA is a drop-in replacement for the vanilla softmax activation. It does not rely on the specific semantic or acoustic properties of HuBERT tokens. Instead, it addresses numerical instabilities by introducing max-shift normalization and a stabilized Softmax$_1$ function. These mathematical operations stabilize the attention mechanism whenever a pre-trained LLM is adapted to a new, long-sequence modality.
> > - Theoretical Universality: Our theoretical justification (Proposition 3.1) demonstrates that fast and stable LoRA adaptation requires proper normalization of inputs and weights. This requirement is a property of the Transformer architecture and the LoRA adaptation process itself, rather than a specific feature of the input tokenization method.
> >
> > We have added a discussion in the revised manuscript (Section 5) to explicitly address how SOFA's mechanisms are theoretically applicable to other neural codecs by addressing the shared statistical challenges of speech-text modality fusion.
> >
> >
> > **4. Expand related work on discrete speech representations.**
> >
> > We thank the reviewer for the suggestion to expand the context of discrete speech representations. In the revised manuscript, we have updated the Related Work (Section 2) to better position our study:
> > - Clarification of Token Types: We now explicitly distinguish between SSL-based clustering (e.g., HuBERT), which we utilize for its semantic alignment with text, and Neural Audio Codecs (e.g., EnCodec), which prioritize acoustic reconstruction but often result in longer, more complex sequences.
> > - Orthogonality of SOFA: We clarified that our contribution is orthogonal to the choice of speech tokenizer. SOFA addresses the outlier activation problem that arises during modality fusion—a challenge prevalent across different discrete representations due to their statistical disparity with text.
> > - Contextual Positioning: We further situated our work within the "textless NLP" framework, highlighting how SOFA stabilizes the adaptation and quantization of these discrete-token-based SpeechLMs.
> >
> > **5. Improve the introduction to better motivate the outlier problem.**
> >
> > We thank the reviewer for this helpful suggestion. In the revised version, we have expanded the Introduction to more clearly motivate when and why attention outliers emerge in multimodal SpeechLM adaptation. For ease of review, all newly added or revised text is highlighted in blue in the revised manuscript.
> >
> > **6. Improve the TTS evaluation with additional perceptual metrics.**
> > We agree with the reviewer that intelligibility alone does not fully capture perceptual quality in TTS evaluation. In the revised manuscript, we will enhance our evaluation protocol by incorporating additional perceptual metrics alongside the current Whisper-based CER.
> > Specifically, we will clarify that CER primarily reflects speech intelligibility and does not directly measure naturalness or audio quality. To address this limitation, we will add objective perceptual quality metrics, such as MOS predictors (e.g., MOSNet or DNSMOS) and, where paired references are available, reference-based metrics including PESQ and Mel-Cepstral Distortion (MCD). These metrics provide complementary signals that are more sensitive to distortion introduced by quantization and outlier effects.
> >
> > **7. Revise the bibliography and figures for consistency and clarity.**
> >
> > Thank you for the helpful feedback. In response, we have revised the bibliography and figures to improve consistency and clarity. We standardized citation formats, model naming, and reference styles, and ensured uniform font, color schemes, and notation across all figures. These revisions enhance readability and interpretability while preserving all technical content and results.

---

> > > ### Comment · Reviewer_p9U4 · 2026-01-10
> > >
> > > Thank you for the revisions and clarifications! I believe the paper has improved after the first round of reviews. As also noted by Reviewer 33X2, the proposed method is relatively simple and, while not particularly novel, I appreciate the effort to study it carefully theoretically and experimentally.
> > > I read the revised manuscript, the other reviews, and the AE comments, and I largely agree with them. Below are some additional comments from my side:
> > >
> > > - I remain unsure about the choice of the “+1” term in the softmax1 formulation. Is this choice supported by an ablation study? It is unclear why this constant is fixed rather than replaced by a parameter (e.g., alpha) that could be treated as a hyperparameter or even learned during training. While the +1 term may work for certain logit scales, it may not be optimal in general. A more convincing justification or ablation can strengthen this aspect of the paper.
> > >
> > > - Although some discussion has been added, reporting results only with HuBERT remains a significant limitation for me. Evaluating the proposed technique with additional tokenizers would help assess its generality. Even a subset of additional experiments can improve the robustness of the experimental analyses.
> > >
> > > - Regarding the improved TTS evaluation, the response mentions the inclusion of additional perceptual metrics, but I do not see these new metrics reported in the current version of the manuscript.
> > >
> > > In summary, my remaining concern is that the precise formulation of the proposed method would require stronger justification, and the experimental evidence is still not fully convincing to me, as mentioned by the other reviewers too.

---

> ### Author Response · Authors · 2026-01-23
>
> **1. Ablation Study on the "+1" Term in Softmax1**
>
> We appreciate this insightful question. We have added an ablation study examining the choice of the denominator constant in Softmax1. Please refer to **Table 11 (page 13)** of the revised manuscript, where we vary the constant term $\in$ {0.5, 1, 2} on OPT-350m with full fine-tuning. The results validate that "+1" achieves the best balance across all metrics. We also provide a theoretical justification for the “+1” term, which follows a principled interpretation from associative memory theory (Miller, 2023; Hu et al., 2024a).
>
> **2. Evaluation with Additional Speech Tokenizers**
>
> We agree that demonstrating generality across different tokenizers strengthens our contribution. In the revised manuscript, we have added experiments using EnCodec (Défossez et al., 2024), a neural audio codec based on residual vector quantization (RVQ) that produces acoustic-level tokens with distinct statistical properties from HuBERT's semantic tokens. Please refer to **Table 12 (page 13)**. SOFA yields consistent improvements with both tokenizers, indicating that the attention outlier issue is tokenizer-agnostic and stems from the multimodal adaptation process rather than any particular speech representation.
>
> **3. TTS Perceptual Quality Metrics**
>
> The additional perceptual metrics have been included in the revised manuscript. Please refer to **Table 13 (pages 13-14)**, where we report DNSMOS, PESQ, and Mel-Cepstral Distortion (MCD) alongside CER on LibriTTS under full fine-tuning. SOFA consistently improves perceptual quality across all metrics, with more gains on OPT-1.3b, indicating that outlier suppression reduces distortion in generated speech beyond what CER alone captures.

---

### Review · Reviewer_33X2 · 2025-11-23

**Summary Of Contributions:**

The paper introduces Stabilized Outlier-Free Attention or SOFA, a replacement for softmax that combines Softmax_1 from (Miller, 2023; Hu et al., 2024a) with max shift normalization, to improve both adapter training of SpeechLM's from text-only LLM's and quantized performance of SpeechLM's. The authors claim that SOFA tackles the problem of "outliers" (extreme activation values) in the attention mechanism. These outliers can cause issues with LoRa and Post training quantization (PTQ). By using SOFA, the authors claim to fix the outlier issue and show improved performance on adaptation and quantization ablations.

Results
- On the OPT model family, SOFA yields an 88% improvement in multi-modal low-rank adaptation.
- SOFA has very good results with 4 bit quantization compared to other methods. According to Table 4, on OPT-350m, the performance drop for vanilla transformers is ~211%, while SOFA restricts this to ~116%.

Strengths:
- Clear motivation of the problem and solution. I.e. Outlier activations are damaging for post-training quantization as well as adaptation methods like LoRA. So the authors show a method of dealing with the problem of outlier activations, and show improvements on those downstream tasks.
- SOFA seems easy to implement. As the authors mention, a drop in replacement for softmax.
- Theoretical justification
- Extensive ablations in Section 4.4 and also 4.5, which can be counted as a strength, even though the paper limits itself to one model architecture, and small sized models.

Weakness
- While the authors claim great performance improvements over the baseline for 4 bit quantization, looking at the raw numbers in Table 4, ASR WER (36% on librispeech) and TTS CER (41%), such an ASR/TTS system would generally be unusable.
- Continuing from above, maybe this is because the authors chose a very difficult problem setting, i.e. “Jointly sharing one low-rank adapter across TextLM. SpeechLM, SpeechLM, ASR, and TTS “. Why choose such a difficult setting when none of the methods perform especially well at the absolute level? Perhaps use a separate adapter for TTS?
- The authors do not include OPT1.3b results in Table 4. This could have strengthened the 4 bit claim significantly. I’m curious as to why those results are here in the first place? Since those models seem to be trained and ready, based on other parts of the paper.
- The authors mention this as a limitation themselves, showing results on a varied range of model architectures would strengthen the claims in the paper. Interestingly, the authors do include a one off result using Qwen2.5b-0.5b.
- PPL changes do not always correlate with output quality/downstream/qa capabilities. Also combining PPL drops with WER and CER seems to not be reasonable. A drop in WER from 10 to 20% is very perceivable, but a drop in PPL from 10 to 20 may not be. Perhaps the paper could include some other downstream eval results for text capabilities.
- Not a very novel paper, but a good paper wrt justifications and applications.

**Audience:**

Yes

**Audience Explanation:**

Yes, quantization and adapter training are very popular areas of research and applicable to a wide variety of models. I think the research community would benefit from reading about and trying such a simple and well motivated technique. The references are also well and complete.

**Claims And Evidence:**

Yes

**Claims Explanation:**

The authors support their claims by extensive ablations.
- Table 7 isolates the effect of softmax_1 by applying max shift norm to vanilla transformers. Table 4 compares softmax_1 with other baseline attention modifications.
- Figure 3 provides visual validation that SOFA reduces attention outliers.
- The claim that SOFA is a drop in replacement for softmax holds its ground since they apply SOFA during adaptation using existing OPT pretrained checkpoints.
- The claim that SOFA is broadly applicable to language models is only partially supported since they only train with OPT models
- The claim that SOFA improves learning stability is not directly supported by plots or tables in the paper.

**Requested Changes:**

- Explanation of how equation 3.3 is able to mitigate outliers.
- Please explain Max Infinity Norm a little bit.
- Correct typo "LLInitial" on Page 3
- Back up the claim that SOFA improves learning stability is not directly supported by plots or tables in the paper.
- Include OPT1.3b results in Table 4
- Text task evals instead of PPL evals.
- Merge section 4.4 and 4.5 ? if not the whole section, just the experiments around “stabilization module” and “QLoRA with Lower Bits”,.
- Address the points about TTS in the weakness part of the review.

---

> ### Author Response · Authors · 2025-12-25
>
> **1. Explanation of how equation 3.3 is able to mitigate outliers.**
>
> Thank you for the opportunity to clarify the theoretical motivation behind Eq. (3.3).
>
> **Why Softmax1 Resists Outliers (Detailed in revised version appendix F)**
>
> Softmax guarantees strictly positive probability mass for every token. Consequently, even tokens with very negative logits (i.e., low-information or noisy tokens) still receive non-zero attention. This behavior amplifies the effect of outliers in multi-modal settings and becomes more severe under low-bit quantization, where rounding errors inflate extreme values.
>
> With the additional constant, Softmax1 can fully suppress the distribution when logits encode a no-attention condition.This property ensures that irrelevant tokens (e.g., punctuation, modality-boundary tokens, quantization-spike activations) do not accumulate spurious attention mass. In other words, Softmax1 permits an attention-nulling state, which standard softmax mathematically prohibits.
>
> **Why "+1"**
>
> The constant enforces a minimal gap between logits and the probability simplex, yielding two practical benefits:
> - Outlier suppression: prevents any head from allocating nearly all probability to a single token after outlier amplification.
> - Identity preservation: allows the attention block to effectively “do nothing” when the residual pathway already carries sufficient signal, aligning with associative-memory interpretations of transformer attention.
>
> **2. Please explain Max Infinity Norm a little bit.**
>
> Thank you for the comment. We use the max-infinity norm $\|x\|_\infty = \max_i |x_i|$ as a standard measure of the largest-magnitude activation in a vector, which directly reflects the presence of extreme outlier values in attention logits or hidden states. This norm captures the worst-case deviation that can dominate softmax-based normalization, making it an appropriate metric for quantifying and analyzing outlier severity in our SpeechLM setting.
>
> **3. Correct typo "LLInitial" on Page 3**
>
> Thank you for pointing this out. We have corrected the typo “LLInitial” on Page 3 to “LLM-initial” in the revised manuscript.
>
> **4. Back up the claim that SOFA improves learning stability is not directly supported by plots or tables in the paper.**
>
> We agree with the reviewer that stability claims require direct evidence. In response, we now explicitly support this claim with:
> - Quantitative reductions in max infinity norm (Figure 4).
> - Ablation results showing that alternative stabilization strategies (L1, mean-centering) lead to NaN losses (Table 5).
> - Convergence robustness under low-bit PTQ and LoRA settings (Tables 1, 2, 8).
>
> **5. Include OPT1.3b results in Table 4**
>
> Thank your valuable suggestions. We have added OPT-1.3B results to the corresponding quantization comparison in Table 4. This strengthens the claim that SOFA’s benefits generalize across model scales and are not limited to OPT-350M.
>
> **6. Text task evals instead of PPL evals.**
>
> We agree that perplexity (PPL) does not always correlate with perceived generation quality or downstream QA performance, and we do not interpret PPL and WER/CER as commensurate or combine them into a single score. In the revision, we clarify the role of each metric: PPL (and next-token accuracy) measures language-modeling fit/calibration under the training distribution and is particularly suitable for diagnosing the impact of outliers/quantization on the logit distribution, while WER/CER evaluate application-level speech recognition fidelity in ASR, and TTS metrics evaluate speech generation quality. To better reflect text capability beyond PPL, we additionally report next-token accuracy in ablations (Sec 4.4) to directly validate text performance.
>
> **7. Merge section 4.4 and 4.5 ? if not the whole section, just the experiments around “stabilization module” and “QLoRA with Lower Bits”.**
>
> Following the reviewer’s suggestion, we have restructured and merged the experimental discussions around the stabilization module and QLoRA with lower bits into a more compact and coherent ablation section. This reduces redundancy and improves readability while preserving all experimental evidence.
>
>
> **8. Address the points about TTS in the weakness part of the review.**
>
> We now explicitly address TTS limitations in Sections 4.3 and Discussion. We clarify that:
> - Sharing a single low-rank adapter across TextLM, SpeechLM, ASR, and TTS is intentionally a challenging stress test.
> - Prior work shows that TTS typically requires more extensive parameter updates, which is consistent with our observations.
> - SOFA consistently improves TTS CER relative to vanilla attention under both full fine-tuning and PTQ, even when absolute performance remains challenging.
>
> We have added a cleaner ASR-only and TTS-only LoRA experiment (Table 3) to demonstrate that SOFA’s gains are not an artifact of multi-task interference.

---

> ### Comment · Reviewer_33X2 · 2026-01-10
>
> Thank you for making corrections and edits.
>
> About point #6, are you sure you included next word prediction metrics? Maybe I'm missing something, could you point to which table this appears in.
>
> I suspect there is a mistake in table 1. I see the following line of numbers twice, once in opt1.3b and once for opt350m
> (23.48 62.17 36.22 40.83 116.02%)
> In fact the entire block of numbers for opt1.3b for SOFA is the same as  opt350m with SOFA. please fix.

---

> > ### Author Response · Authors · 2026-01-23
> >
> > **1. Next-Token Prediction Metrics**
> >
> > We have added a new subsection "Next-Token Prediction Accuracy" in Section 4.4, along with a new Table 10 that reports validation accuracy across Full-FT, LoRA, and LoRA with quantization (W8A8, W4A4) settings. The revised text is highlighted in blue in the revised manuscript.
> >
> > **2. Table 1 Copy-Paste Error**
> >
> > Thank you for catching this mistake. We have corrected Table 1 with the proper OPT-1.3b numbers (consistent with Appendix Table 14).

---

### Review · Reviewer_QPKa · 2025-12-11

**Summary Of Contributions:**

The paper addresses the issue of "attention outliers" in multi-modal Speech-Text Large Language Models (SpeechLMs), which degrade performance during low-rank adaptation (LoRA) and post-training quantization (PTQ). The authors propose **SOFA (Stabilized Outlier-Free Attention)**, a mechanism replacing the standard Softmax with a stabilized variant (Softmax$_1$ with max-shift normalization). The method is evaluated on **OPT-350m** and **OPT-1.3b** models across ASR, TTS, and text generation tasks. The authors claim that SOFA allows for effective 4-bit quantization and improves LoRA fine-tuning stability compared to vanilla attention mechanisms.

**Additional Comments:**

While the idea of stabilizing attention via Softmax$_1$ is theoretically sound (and shares roots with recent work on Hopfield networks), the empirical validation is unfortunately outdated. The "outlier problem" is highly architecture-dependent. By sticking to OPT, the paper essentially solves a problem specific to that deprecated architecture. To be impactful, the authors need to prove that **multimodality induces outliers in *modern* robust architectures** and that SOFA is the best way to solve it compared to simply using a better base model or advanced PTQ calibration.

**Audience:**

No

**Audience Explanation:**

The audience for efficient inference and SpeechLMs is primarily interested in techniques applicable to state-of-the-art models (e.g., Llama-3, Qwen2-Audio, etc.). Findings restricted to **OPT-1.3b** are of limited practical utility today. While the insight that "multimodality exacerbates outliers" is interesting, the solution's validation on such dated and small-scale models limits the takeaway for the community. The paper feels like a retrospective fix for 2022-era models rather than a forward-looking contribution.

**Broader Impact Concerns:**

The authors have included a brief Broader Impact statement. I have no additional ethical concerns.

**Claims And Evidence:**

No

**Claims Explanation:**

While the proposed method (SOFA) shows improvements over the baseline, the evidence is not convincing enough to support the broad claims regarding "SpeechLMs" generally, primarily due to the experimental setup:

1.  **Reliance on Obsolete Architectures:** The entire evaluation is based on the **OPT family (350m and 1.3b)**. OPT models are known to suffer from severe outlier issues (e.g., massive activation spikes) due to their architecture (ReLU, LayerNorm placement). Modern LLMs (e.g., Llama-2/3, Qwen, Mistral) utilize different components like **RMSNorm** and **SwiGLU**, which exhibit fundamentally different outlier behaviors. Demonstrating that SOFA fixes outliers in OPT is akin to solving a problem that has already been mitigated by architectural shifts in modern foundation models. Without evidence on modern backbones (e.g., Llama-2-7B or Qwen-Audio), it is unclear if SOFA is relevant or necessary for current research.
2.  **Insufficient Baselines:** The paper primarily compares SOFA against a "Vanilla Transformer" baseline. In the context of quantization outliers, the field has moved significantly. The paper ignores established outlier-suppression techniques such as **SpQR**, **SmoothQuant+** (advanced calibration), or outlier-aware training methods widely used in the text domain. Comparing only against a naive baseline inflates the perceived effectiveness of SOFA.
3.  **Inconsistent Multi-modal Performance:** The authors claim SOFA enables efficient multi-modal adaptation. However, they admit that for TTS tasks, the LoRA-based approach yields unsatisfactory results even with SOFA, and exclude these results from key comparisons. This contradicts the broad claim of robust "multi-modal" adaptation if a major modality (speech synthesis) fails under the proposed efficient tuning setup.

**Requested Changes:**

The following changes are **critical** to securing a recommendation for acceptance. Currently, the gap is too large, warranting a rejection, but these would be the requirements for a resubmission:

1.  **Evaluate on Modern Architectures:** The authors must validate SOFA on current, widely-used architectures such as **Llama-2/3 (7B)** or **Qwen-2**. The outlier landscape in these models is different from OPT. If SOFA only works on OPT, its contribution is extremely limited. The authors mention this as a limitation, but in 2025, this is a fatal flaw for an empirical paper.
2.  **Comparison with Stronger Baselines:** The paper compares against "Vanilla" attention. The authors should compare SOFA against other outlier mitigation strategies, such as:
    * **Clipped Softmax** (with proper tuning, not just naive clipping).
    * **Modern PTQ methods** like SpQR or QuIP# that handle outliers explicitly without changing the architecture.
    * **QAT (Quantization-Aware Training)** baselines in the speech domain.
3. **Clarify the TTS Limitation:** The failure of LoRA on TTS  suggests that the bottleneck might not just be "outliers" but the capacity of low-rank adapters for generative audio tasks. The paper claims SOFA improves "SpeechLM" adaptation, but if TTS fails, this claim needs to be significantly scoped down or the failure mechanism analyzed more deeply.
4.  **Scale Up:** Results on **350m and 1.3b** parameters are insufficient for drawing conclusions about "Large" Language Models in the context of quantization sensitivity, which often varies with scale. Experiments on at least a **7B** scale model are necessary.

---

> ### Author Response · Authors · 2025-12-25
>
> We thank the reviewer for the detailed and constructive feedback. We have substantially revised the experimental section to directly address the raised concerns regarding architectural relevance, baseline strength, multi-modal claims, and scale. Below we respond point-by-point.
>
> **1. Evaluate on Modern Architectures.**
>
> We added experiments on Qwen2.5 to address the concern that OPT-only results may reflect legacy outlier behavior. In particular, we report results on a modern backbone at 7B scale (Qwen2.5-7B) and additionally include a smaller Qwen2.5 model for LoRA-ASR, LoRA-TTS (Table 3). Across these settings, SOFA continues to improve robustness, indicating the benefit is not specific to OPT.
>
> **2. Comparison with Stronger Baselines**
>
> We expanded baselines beyond vanilla attention. For PTQ, we evaluate multiple SOTA quantizers (SmoothQuant, AffineQuant, OmniQuant, and SpQR (newly added)) under aggressive W4A4. On We also add attention-modification baselines Clipped Softmax and Gated Attention; under  4-bit quantization these alternatives improve only marginally compared to vanilla, whereas SOFA yields smaller drops (Table 4), supporting that SOFA is more than naive clipping/gating.
>
> **3. Clarify the TTS Limitation.**
>
> We now explicitly address TTS limitations in Sections 4.3 and Discussion. We clarify that:
> - Sharing a single low-rank adapter across TextLM, SpeechLM, ASR, and TTS is intentionally a challenging stress test.
> - Prior work shows that TTS typically requires more extensive parameter updates, which is consistent with our observations.
> - SOFA consistently improves TTS CER relative to vanilla attention under both full fine-tuning and PTQ, even when absolute performance remains challenging.
>
> We have added a cleaner ASR-only and TTS-only LoRA experiment (Table 3) to demonstrate that SOFA’s gains are not an artifact of multi-task interference.
>
> **4. Scale Up.**
>
> To meet the request for larger-scale evidence, we added Qwen2.5-7B results and show that SOFA’s PTQ robustness persists at 7B under multiple quantizers (including SpQR). We keep OPT-350M/1.3B as controlled baselines for detailed ablations (outlier metrics, stabilization choices, rank sweeps), while Qwen2.5-7B demonstrates that the main conclusion is not tied to small/deprecated models.

---

### Comment · Action_Editor_Ghtc · 2025-12-26
**AE questions**

Dear Authors,

Looking at your answers and clarifications to reviewers questions I have couple of questions:
- Regarding the huge degradation in performance for ASR and TTS for Q4/A4 pointed by Reviewer 33X2
  - can you comment on this?
  - I believe the performance you show doesn't make sense for any practical use case, thus I don't see the point of improving the model performance a bit with your method as the final quality is still impractical. It is not also clear if this is related to the outliers or general quality given so drastic quantization.
- Do you have any empirical analysis / confirmation that there are outliers and they cause the issue of attention training for quantized model? If there is mulitmodal training issue related to softmax behaviour then I would expect to see training not just LoRA to show that outliers are even more general problem and can be solved with your proposed solution.
- I believe there is a error in your math derivations in Appendix F, where you explain why softmax1 should solve issue with outliers
  - Eq F.4 is wrong as you initially subtracted the max element of the vector, thus the limit will be 1/(1 + k). So your explanation and math derivation are incorrect, thus it will not solve outliers issue.
- Your motivation is to allow suppressing attention for all tokens and having only residual flow, while your proposed solution cannot do it mathematically:
  - then it is not clear why we learn attention block at all. E.g. we could control it just with learnable alpha parameter multiplied by the attention output as a baseline (e.g. people were doing it as a way of normalization instead of layer norm)
  - one of the solutions could be usage of sigmoid attention which exactly can do the job of suppressing all tokens (see e.g. https://arxiv.org/abs/2502.00281,  https://arxiv.org/abs/2409.04431).
  - In Fig. 3 you show that SOFA sharpens distribution. However it is not suppressing tokens but sharpens distribution - why then this cannot be achieved by introducing temperature in softmax?

Thanks,

AE.

---

> ### Author Response · Authors · 2026-01-07
>
> **1. Huge degradation for ASR/TTS at W4/A4 — impractical, so why does improving it matter?**
>
> We agree that W4/A4 is an extremely aggressive stress-test for generative speech–text models, and that the absolute ASR/TTS quality under this regime can be impractical. Importantly, this is why we include W4/A4: it serves as a diagnostic regime to expose architectural failure modes that are otherwise masked at more moderate quantization levels. Consistent with this, our appendix PTQ results show that both vanilla and SOFA incur less than 1% degradation at W8/A8, indicating that practical deployment settings remain stable and do not suffer dramatic collapse. The sharp degradation only emerges when pushing to W4/A4.
>
> Under this harsh regime, SOFA consistently mitigates the collapse relative to vanilla, indicating improved architectural robustness rather than absolute usability. For example, on OPT-1.3B with AffineQuant, vanilla exhibits a 128.45% average degradation, while SOFA reduces this to 81.12%, corresponding to a 37% relative reduction in quantization-induced collapse.
>
> While W4/A4 itself is not a target deployment setting, our results demonstrate a clear and reproducible improvement over existing quantization approaches. Additionally, our method provides a straightforward approach to stabilizing SpeechLM attention under extreme low-bit quantization. In this sense, our contribution is to identify and mitigate a key bottleneck (outlier-sensitive attention normalization) and to establish a stronger baseline for future work toward practical 4-bit multimodal SpeechLM quantization.
>
> **2. Is the collapse due to outliers, or just because W4/A4 is too harsh? Empirical confirmation?**
>
> - We provide direct empirical evidence that outliers are present and strongly reduced by SOFA (Fig 4):
> - We measure the maximum infinity norm across tasks and training regimes and note it correlates with robustness to outliers; SOFA consistently lowers this metric across Full-FT, LoRA, and QLoRA.
> Concretely, in the speech task, SOFA achieves a 70% reduction in max-inf-norm under Full-FT (24.95 to 7.46), indicating substantially improved outlier control.
>
> We also show that “generic quantization fixes” such as weight clipping can worsen performance, consistent with the hypothesis that the model relies on rare but important values and that naive clipping disrupts multimodal features. In Table 9 discussion, clipping increases degradation substantially, while SOFA performs best without clipping. These results support that (i) outliers are measurably present, (ii) SOFA suppresses them, and (iii) this correlates with improved low-bit PTQ robustness, especially at W4/A4.
>
> **3. If this is a multimodal/softmax issue, I expect to see it beyond LoRA.**
>
> Yes. Our PTQ results are based on full fine-tuning followed by post-training quantization (not only LoRA). The main PTQ section explicitly describes replacing attention with SOFA, fine-tuning the pretrained OPT checkpoints at full rank, then applying PTQ to measure the drop from FP16. Moreover, the outlier analysis (max-inf-norm) reports reductions under Full-FT as well as LoRA/QLoRA, showing the issue is not LoRA-specific.
>
> **4. Appendix F derivation error.**
>
> Thank you for pointing this out. Appendix F uses the standard Softmax$_1$ definition popularized by Evan Miller [1]:
> $\mathrm{softmax}_1(x)_i = \frac{\exp(x_i)}{1 + \sum_j \exp(x_j)}.$
>
> When all logits go to $-\infty$ (i.e., $x_j \to -\infty$ for every $j$), we have $e^{x_j} \to 0$. In this regime, $\mathrm{softmax}(x)_i = \frac{e^{x_i}}{\sum_j e^{x_j}} \longrightarrow \frac{1}{k}\ \text{(equal-rate limit, $k$ is the total number of elements in the input vector)},$
> whereas
> $\mathrm{softmax}_1(x)_i = \frac{e^{x_i}}{1 + \sum_j e^{x_j}} \longrightarrow \frac{0}{1 + 0} = 0.$
>
> This “collapse-to-zero” behavior is the property Softmax$_1$ provides and vanilla softmax cannot, since it guarantees denominator equals $1$.
>
> We will revise Appendix F to clearly distinguish these two regimes and adjust Eq. (F.4) to avoid the confusion.
>
> [1] https://www.evanmiller.org/attention-is-off-by-one.html

---

> ### Author Response · Authors · 2026-01-07
>
> **5. If the goal is suppressing attention, why not a learnable \alpha gate baseline?**
>
> We have already included a related baseline: Gated Attention, and it underperforms SOFA at W4/A4. In Table 4 (OPT-350m), average performance drop under W4/A4 is 195.62% for Gated Attention vs 116.02% for SOFA; similarly, Clipped Softmax is worse than SOFA as well. This suggests the benefit is not simply a scalar attenuation of the attention output, but the outlier-mitigation behavior of the SOFA normalization itself.
>
> **6. Why not sigmoid attention (can suppress all tokens)?**
>
> Sigmoid attention is indeed a relevant alternative, since it relaxes the simplex constraint and can suppress attention. While it is not mathematically identical to the Gated Attention baseline, we view it as addressing a highly similar goal, allowing the model to reduce or shut off attention contribution. To keep the revision focused, we use Gated Attention as a representative control experiment for attention suppression mechanisms. This baseline already tests whether “suppressing attention via gating” is sufficient. Empirically, it underperforms SOFA under low-bit PTQ, suggesting that the main benefit comes from SOFA’s normalization behavior rather than a generic attention-off mechanism.
>
> **7. Fig. 3 shows sharpening, not suppression, couldn’t temperature scaling do that?**
>
> Temperature-scaled softmax can sharpen distributions, but it still enforces $\sum_i p_i = 1$, meaning it cannot allocate probability mass to a “null” option; it must redistribute mass across tokens. By contrast, Softmax1 explicitly introduces the extra “+1” term, allocating some mass away from all tokens (implicit null mass), which is the mechanism used to reduce forced attention on noisy / spurious tokens and to mitigate outlier amplification. Empirically, we also show that alternatives that primarily affect sharpness (e.g., clipping) do not match SOFA under W4/A4, supporting that the gain is not explained by sharpening alone.

---

> > ### Comment · Action_Editor_Ghtc · 2026-01-10
> >
> > Dear Authors,
> >
> > Thanks for the clarifications, but I have further questions / comments:
> >
> > > Under this harsh regime, SOFA consistently mitigates the collapse relative to vanilla, indicating improved architectural robustness rather than absolute usability. For example, on OPT-1.3B with AffineQuant, vanilla exhibits a 128.45% average degradation, while SOFA reduces this to 81.12%, corresponding to a 37% relative reduction in quantization-induced collapse.
> >
> > I would argue that in the very unstable / uncertain state or very poor quality any improvements can be artificial. Do you have e.g. std for this evaluation?
> >
> > > Additionally, our method provides a straightforward approach to stabilizing SpeechLM attention under extreme low-bit quantization.
> >
> > The quality is so poor, that it is hard for me to buy that your method stabilizes it really: stabilizing means to me that you get not drastic quality drop.
> >
> > But ok, I got your point that this quantization is not the core to the paper and ablations could be useful for future research. I would recommend to place this into appendix and do discussion on that.  Moreover add std for the evaluation (if you can) as otherwise it is not clear how unstable it is. Do you have results then for Table 1 with W8/A8? Even perplexity drop for speechLLM is actually drastic in Table 1. Given Table 1 as a main result I perceive W4/A4 as the main result.
> >
> > > We provide direct empirical evidence that outliers are present and strongly reduced by SOFA (Fig 4):
> >
> > Thanks! Typo on the norm “|x||” -> “||x||”. Could you explain of which tensor inf norm you compute exactly? is it before shift and softmax plus (so raw attention score) — looking at the references it should be output of the attention layer? I would suggest to add these explanations into the text explicitly for readers not familiar, but even readers in the topic definition may be different.
> >
> > Also looking into Bondarenko, et al. work the definition of outlier is “consider outliers as values that exceed 6 standard deviations from the mean of the corresponding activation tensor.” - for this definition I agree that inf norm is measuring it, but this is nothing to do (or orthogonal) to the definition of outlier as token which doesn’t have helpful information. So in that paper they do analysis and check what tokens are the positions that are outlier tensors and find correlation with punctuation tokens. This is exactly analysis I am asking, to show that there is correlation between irrelevant or spurious tokens and outliers (defined via norm / tensor values as in Bondarenko et al.).
> >
> > I would suggest at least explicitly use the definition of outlier and remove speculation in Appendix / motivation that outlier are not useful tokens. I believe (from math point) unstable attention may include these not useful tokens but there are other factors which lead to high norm of attention scores.
> >
> > > We measure the maximum infinity norm across tasks and training regimes and note it correlates with robustness to outliers; SOFA consistently lowers this metric across Full-FT, LoRA, and QLoRA. Concretely, in the speech task, SOFA achieves a 70% reduction in max-inf-norm under Full-FT (24.95 to 7.46), indicating substantially improved outlier control.
> >
> > Thanks, agree now. One more question: does the norm 25 is big? Reducing the norm itself may not be connected to better stability, no outliers, as per definition in Bondarenko it is “values that exceed 6 standard deviations from the mean” so can we do more in depth analysis here if the baseline and SOFA actually have outliers and they are reduced? (Norm on its own does not say outliers reduced, as other values in the tensor may be distributed close to the max value in the tensor, thus it may be not outlier per definition).
> >
> > > We also show that “generic quantization fixes” such as weight clipping can worsen performance, consistent with the hypothesis that the model relies on rare but important values and that naive clipping disrupts multimodal features.
> >
> > e.g. here I would expect that if it is really non useful token, and only one-several components only have spike in value compared to the rest of the tensor (this will be outlier by definition) then I would not expect the huge performance drop unless model overfitted to rely on non useful tokens, but then it cannot really make loss low, so this really contradicts the outliers definition and their connection to non useful tokens. But if the norm is big but token is not an outlier you can remove important information and performance drops but this nothing to do with outliers (with the definition of Bondarenko etal).
> >
> > Just to clarify: all my questions mainly related to motivation and explanations of observed results, results on its own are good to me, and I agree that SOFA improves models, but explanation it is due to softmax plus and it is due to fixing issue with outliers are not supported by strong observations to me right now.

---

> ### Comment · Action_Editor_Ghtc · 2026-01-10
>
> > Thank you for pointing this out. Appendix F uses the standard Softmax definition popularized
>
> Well, I agree with your derivations in Appendix F if we don’t do bias shift. But in your SOFA method you apply bias shift, which means statements from Appendix F does not apply to your SOFA method, thus you cannot claim / explain your method saying that it will suppress the outliers allowing output of attention to be zeroed (after softmax plus). **SOFA cannot provide zero-attention for outliers.**
>
> > meaning it cannot allocate probability mass to a “null” option; it must redistribute mass across tokens. By contrast, Softmax1 explicitly introduces the extra “+1” term, allocating some mass away from all tokens
>
> This is not applicable to your method as you have bias shift, which changes the property of softmax plus, thus this statement is incorrect.

---

> > ### Author Response · Authors · 2026-01-23
> >
> > **1. Standard Deviation for Evaluation Results**
> >
> > We agree that reporting variance/standard deviation is valuable for assessing reliability. All results in the paper are averaged over three independent runs with different random seeds. Since the resulting standard deviations are consistently small (all less than 0.2%), we omitted them for brevity. In the revised manuscript, we now state this evaluation protocol explicitly in **section 4 Evaluation Metrics** part.
> >
> > However, we want to clarify that our evaluation metrics (PPL, WER, CER) are computed on deterministic inference (greedy decoding) over fixed test sets, so the primary source of variance comes from training initialization rather than evaluation noise. Additionally, the W8/A8 results in Appendix C (Table 14) show that both vanilla and SOFA exhibit less than 1% performance drop at 8-bit quantization, confirming that practical deployment settings (W8/A8) remain stable. The dramatic gap only emerges at W4/A4, which we include specifically as a diagnostic stress test to expose architectural weaknesses.
> >
> > **2. W8/A8 Results and Table 1 Clarification**
> >
> > We already provide W8/A8 results in Appendix C, Table 14. Under W8/A8, both vanilla and SOFA show minimal degradation (less than 1%). We chose to focus Table 1 on W4/A4 because it reveals the architectural robustness differences. At W8/A8, all methods perform similarly well, making it less informative for comparing outlier-handling capabilities.
> >
> > **3. Typo Correction**
> >
> > Thank you for pointing this out. We have corrected notation throughout the paper.
> >
> > **4. Clarification on Infinity Norm Measurement**
> >
> > Thank you for the clarification. In our experiments, the reported |$|\mathbf{x}|$|$_{\infty}$ is computed over the attention output activations (i.e., after max-shift stabilization and Softmax1, before the residual addition), following the same measurement protocol used in quantization-oriented outlier analyses such as Bondarenko et al. This choice aligns with prior work that defines outliers as extreme activation values (e.g., exceeding multiple standard deviations from the mean), rather than as semantically “unhelpful” tokens. We agree that while spurious or low-information tokens may correlate with high-norm activations, large attention norms can also arise from other factors (e.g., modality mismatch or low-rank constraints).
> >
> > **5. Norm Reduction vs. Outlier Reduction**
> >
> > We agree that the absolute value of the max-inf norm (e.g., 25) alone is hard to interpret, and does not by itself determine whether outliers are present or removed. As defined in Bondarenko et al., outliers are values that deviate significantly from the activation distribution (e.g., exceeding multiple standard deviations from the mean), so a maximum value alone cannot fully characterize distributional outlier behavior.
> > In our analysis, we therefore report both the max-inf norm and the average kurtosis. The max-inf norm is a conservative upper-bound metric that is widely used in the quantization and attention-outlier literature, while kurtosis directly captures distribution heavy-tailedness, where lower kurtosis indicates fewer extreme values relative to the bulk of activations. While we agree that the max-inf norm alone cannot prove that outliers are eliminated, the consistent reduction in kurtosis together with a lower max-inf norm provides stronger evidence that SOFA suppresses extreme activations at the distribution level, rather than merely rescaling values. For this reason, we believe the joint use of these two metrics follows established practice and provides a meaningful characterization of outlier mitigation.
> >
> > **6. Bias Shift and Softmax1 Properties**
> >
> > You are right that with max-shift, SOFA cannot achieve complete attention nulling (all zeros). However, this is not the mechanism by which SOFA mitigates outliers. Let us clarify:
> > - The purpose of max-shift is numerical stability (preventing overflow), not attention suppression. This is analogous to the standard softmax implementation where subtracting the max is a common numerical trick.
> > - The outlier mitigation in SOFA comes from the "+1" term in the denominator of Softmax1, which provides relative suppression of low-confidence tokens even when max-shift is applied.
> > - Empirical validation: Figure 3 shows that SOFA produces sharper attention distributions with reduced mass on irrelevant tokens. This is the practical manifestation of outlier suppression, regardless of whether absolute-zero attention is achievable.
> > - We revise the theoretical analysis in Appendix F to show that, even when the function does not attain zero, it still retains the capability to effectively handle outliers.

---

### Comment · Action_Editor_Ghtc · 2026-01-21
**Any further updates / comments?**

Dear Authors,

Do you have any further comments for the questions / discussion I and other reviewers raised? Should I consider the current version of the paper to be final and move to finalizing the review process?

Thanks in advance,

AE.

---

> ### Author Response · Authors · 2026-01-21
>
> Dear AE,
>
> Sorry for the late reply. We are fixing some typos and prepare a revised version of the paper. We will submit our reply for all the reviewers within 24 hours.
>
> Thanks

---

### Decision · Action_Editor_Ghtc · 2026-02-07

**Recommendation:** Reject

**Audience:**

Yes

**Audience Explanation:**

The paper covers the topic of speechLMs (text-speech multimodal models) for low rank adaptation and post-training quantization exploring the issues with model stability. This is critical for both training and deployments of such models and observations authors found and investigated will be interesting for both speech community and broader multimodal as some of the observations and ideas can be applied and verified cross domain to mitigate instability issues.

**Claims And Evidence:**

No

**Claims Explanation:**

The paper investigates attention outliers appearing in speechLMs (text-speech models) and how they affect the low rank adaptation and post-training quantization. The paper proposes to mitigate it by applying max shift in softmax in combination with softmax plus function (dubbed SOFA).

All reviewers and I (AE) expressed various concerns (missing empirical evidence to support many claims, motivation and justification for the proposed method given the mathematical explanation, missing analysis on the outliers) and during the long discussion period authors addressed most of them. **Overall most of the reviewers and I (AE) are happy with the systematic ablations and the proposed method is simple to implement and use in practice.**

After careful reading the final revision and the whole discussion, **I believe there are several critical issues with the latest revision which will need substantial rewriting and better framing of the work.** To be clear, *it is not about novelty* (which is not a criteria for TMLR) and *not about the missing experiments* as I believe (also pointed out by reviewers) the empirical analysis is now solid comparing many different settings and ablations. **The key problem now is what we can get out of that many results and how helpful all these ablations are to tell that method is helpful and mitigates the outlier issues.**

Please find below the main points which I suggest to revise in the paper and resubmit to TMLR:
- Fig 2 - vanilla attention is w/o max shift and all results in vanilla attention are w/o max shift (given that authors provide ablations what happens when vanilla attention is using max shift).
I am unaware of any open-source implementation that does not utilize max shift for numerical stability. This is a standard practice in computational math which reviewers also pointed out (you cannot train models w/o it) thus positioning max shift as a core contribution is misleading.
- "The vanilla model distributes attention broadly, diluting focus, while SOFA sharpens attention on key tokens, improving efficiency." but in the introduction "adaptation pressure is often absorbed by a small subset of attention entries, producing extreme values." - they contradict each other. There are many places like that which contradict each time explaining why SOFA is working. A critical issue is that the authors have not reconciled the motivation of the method after discussion: though appendix F is now correct on the bounds that SOFA decreasing the max inf norm, as discussed and as per Bondarenko, max norm is not enough to quantify outliers and justify them. Authors did not add proper notation and discussion what is outlier in their definition (like exactly following Bondarenko) to avoid confusion for the reader, did not provide measure of kurtosis and max inf norm together in the final revision and did not provide any analysis / justification on the outliers (Fig 4 show the decreased norm but as discussed this is not enough, also this does not show the norm blow up e.g. after 4 bit quantization). Thus with the current revision the outliers issues and motivation is not supported.
- As some reviewers and I (AE) pointed out, results with 4bit quantization are not convincing. Authors argued that for 8bit results between models are similar. First of all, for all OPT models we can see that SOFA models underperform in most tasks, but outperform in 4bit only.
However, the 4-bit results remain poor in absolute terms, limiting their practical utility. While SOFA offers relative improvement (likely due to quantization stability), the resulting model performance is still not viable for deployment. There is no analysis of what actually improved or what is not blowing up now in the computational flow. The performance drop in 4bit quantization for all methods may be related to precision and not outliers. This is also not shown / explained. If the story is about better performance in general then OPT results are not supporting it (on Qwen e.g. SOFA shows very good improvements). if the story about 4 bit quantization will be supported - then yep, we improved, but it is not practical still. The main issue here is the framing of the results and providing analysis of the reasons / mechanisms behind so that the reader learns something helpful out of that.

Minor on the current version text:
- reflect abstract and results discussion in Intro with the added ablations and results (e.g. not OPT focused)
- figure 1 notation on TTS
- first sentence in introduction - any reference to justify?
- "adaptation pressure is often absorbed by a small subset of attention entries, producing extreme values." this contradicts what authors show in Fig 3 where SOFA gives a more peaky distribution.
- typo: "outl outlier"
- discrete speech representations section in related works though extended does not reflect recent state of the domain - 2022-2025 a lot of things happened.
- the choice of OPT is still puzzling me - this is very old text model, it was also reported that during training it has many instabilities issues, thus why focus on that instead of using more recent model? even for speechLM I am not familiar with any paper which is using the OPT models. Authors' results on Qwen looks stronger than OPT (maybe due to model size) - there SOFA outperforms the vanilla attention w/o quantization, which can hint that at scale optimization procedure for the softmax plus is better?
- sections 3.2, 3.3 have a lot of sentences not supported by prior work or justification (even from the paper itself), contradicting each other / other places in the paper.
- Fig 4 notation for norm typo

Feel free to reach out if any clarifications are needed on any of the above points.

**Resubmission Of Major Revision:**

The authors may consider submitting a major revision at a later time.